# Impact of urban geology on model simulations of shallow groundwater levels and flow paths

Ane LaBianca[1,2], Mette H. Mortensen[1], Peter Sandersen[1], Torben O. Sonnenborg[1], Karsten H. Jensen[2], Jacob Kidmose[1]

[1]Geological Survey of Denmark and Greenland (GEUS), Copenhagen, Denmark
[2]Department of Geosciences and Natural Resource Management, University of Copenhagen, Copenhagen, Denmark

*Correspondence to*: Ane LaBianca (ala@geus.dk)

**Abstract.** This study examines the impact of urban geology and spatial discretization on the simulation of shallow groundwater levels and flow paths at the city scale. The study uses an integrated hydrological model based on the MIKE SHE code that couples surface water and 3D groundwater simulations with a leaky sewer system. The effect of geological configuration was analyzed by applying three geological models to an otherwise identical hydrological model. The effect of spatial discretization was examined by using two different horizontal discretizations for the hydrological models, respectively 50 m and 10 m. The impact of the geological configuration and spatial discretization was analyzed based on model calibration, simulations of high-water levels, and particle tracking. The results show that a representation of the subsurface infrastructure, and near-terrain soil types, in the geological model impacts the simulation of the high-water levels when the hydrological model is simulated in 10 m discretization. This was detectable even though the difference between the geological models only occurs in 7% of the volume of the geological models. When the hydrological model was run in 50 m horizontal discretization, the impact of the urban geology on the high-water levels was smeared out. Results from particle tracking show that representing the subsurface infrastructure in the hydrological model changed the particles' flow paths and travel time to sinks, both in the 50 m and 10 m horizontal discretization of the hydrological model. It caused less recharge to deeper aquifers and increased the percentage of particles flowing to saturated zone drains and leaky sewer pipes. In conclusion, the results indicate that even though the subsurface infrastructure and fill material only occupy a small fraction of the shallow geology, it affects the simulation of local water levels and substantially alter the flow paths. The comparison of the spatial discretization demonstrates that to simulate this effect the spatial discretization needs to be of a scale that represents the local variability of the shallow urban geology.

## 1 Introduction

As more than half of the world's population lives in urban areas and urbanization globally continues to increase (United Nations, 2018), urban water resources receive increasing attention (McGrane, 2016; Lundy and Wade, 2011; Mitchell, 2006; Farr et al., 2017; Birks et al., 2013). Cities are hydrologically complex, because of interactions between built structures, water infrastructures, such as pumps, drainage, sewers and water pipes, and the natural hydrological system where surface

and subsurface processes occur at various spatial and temporal scales (Salvadore et al., 2015; Han et al., 2017; Kidmose et al., 2015; Fletcher et al., 2013; Tubau et al., 2017; Vázquez-Suñé et al., 2016).

In urban areas, shallow geology is more extensively moderated than in rural areas, both in its physical extent and in the temporal frequency of alteration. This is a consequence of the high population density, which generates the development of high concentrations of buildings and physical infrastructure, as well as transport and utility networks (Attard et al., 2016b; Lerner, 1990, 2002). This shallow geology which is highly affected by man is also referred to as the anthropogenic layer (Mielby and Sandersen, 2017; Ford et al., 2014). The anthropogenic layer can vary in extent and composition within a few meters. Furthermore, the composition and properties of the anthropogenic layer change frequently in urban areas because of the reconstruction of the land surface, maintenance of existing subsurface installations, or new underground constructions (Salvadore et al., 2015; Fletcher et al., 2013; Hibbs and Sharp, 2012; Berthier et al., 2004; Ford et al., 2014). Mielby and Sandersen (2017) argue that it is important to know the city's history to quantify the anthropogenic layer and suggest that different geological modeling approaches are required for the anthropogenic material and the underlying geological sediments respectively.

Previous studies have highlighted that the hydraulic characteristics of the anthropogenic layer can cause preferential flow pathways (Salvadore et al., 2015; Fletcher et al., 2013; Berthier et al., 2004) and that underground structures may act as obstacles to flow (Hibbs and Sharp, 2012; Lerner, 1990, 2002; Pophillat et al., 2022) and increase mixing of shallow and deep aquifers (Attard et al., 2016b, 2017). Yet, only a few studies have considered the impact of anthropogenic modifications on the groundwater flow when modeling urban hydrogeology at the city scale (Berthier et al., 2004; Attard et al., 2017). Berthier et al. (2004) considered the anthropogenic modifications of soils in a 2D numerical model setup. They found that the soil hydraulic conductivities have a significant impact on water table level and runoff from drained layers. However, the model was only applied to a small area (0.05 km$^2$) and neglected evapotranspiration and infiltration on paved surfaces. Moreover, often urban hydrogeological models mainly focus on the impact of the subsurface close to a specific construction site (Mielby and Sandersen, 2017; Laursen and Linderberg, 2017; Attard et al., 2016b; Troldborg et al., 2021). Yet, studies by Attard et al. (2016b, 2017), Boukhemacha et al. (2015), and Epting et al. (2008) show examples of studies that use numerical modeling to quantify the cumulative impact and interaction of multiple underground structures on the groundwater flow. These studies found that urbanization impacts the water balance and flow systems both at local and larger scales. However, they mainly focused on obstacles to the flow and did not consider the entire local heterogeneity of the subsurface properties (Hibbs and Sharp, 2012; Attard et al., 2016b, a, 2017; Locatelli et al., 2017).

Comprehensive numerical groundwater models require detailed geological input. Salvadore et al. (2015), Hutchins et al. (2017) and Mielby and Sandersen (2017) suggest that a detailed spatial description of the landcover and the geological settings, down to a scale of a few meters, is required for urban hydrological modeling. This increases the effort needed to compile and process large amounts of data compared to traditional hydrogeological modeling, including the temporal changes of the urban surface and subsurface (Salvadore et al., 2015; Hutchins et al., 2017). Models of shallow urban geology thus require data that is normally not used in hydrogeological models, such as the location of the subsurface infrastructure

and buildings, and descriptions of back-fill material. Growing urbanization makes it increasingly difficult to survey the urban subsurface (Petrosino et al., 2021; Culshaw and Price, 2011; Mielby and Sandersen, 2017), and high spatial variability of the subsurface and sparse data introduce high uncertainty to the modeling (Salvadore et al., 2015). Nevertheless, studies by Andersen et al. (2020), Mielby and Sandersen (2017) and Vázquez-Suñé et al. (2016) have introduced methods of compiling subsurface data from various sources and developing 3D geological models of the urban subsurface.

Laursen and Linderberg (2017) and Mielby and Henriksen (2020) argue that integrated models of the surface and subsurface processes at the city scale, which can give reliable estimations of the consequences of potential changes, are needed for effective urban water resources management. Current integrated surface-subsurface modeling of urban hydrology however often simplifies the description of the subsurface components (Pophillat et al., 2021). Yet, while simple models might describe the impact of the subsurface on the surface hydrology acceptably, processes and interactions affecting the water

cycle and the subsurface flow system in the presence of a shallow groundwater table may be neglected (Pophillat et al., 2021; Attard et al., 2016c). Common limitations of integrated modeling are data shortage, undescribed processes, and parameterization, which all are sources of uncertainty (Salvadore et al., 2015; Schirmer et al., 2013; Pophillat et al., 2021; Fletcher et al., 2013).

Models of the urban hydrological system at the city scale are typically challenged by the presence of leaking water pipes or

sewers that cause unintended groundwater recharge or drainage. The location and to which degree the pipes are leaking are often poorly documented (Lerner, 2002; Yang et al., 1999; Hibbs and Sharp, 2012; Tubau et al., 2017; Vázquez-Suñé et al., 2010), as are local climatic and hydrological observations (Fletcher et al., 2013; Salvadore et al., 2015; Hutchins et al., 2017; McGrane, 2016). Yet, Hutchins et al. (2017) highlighted that the advances in monitoring techniques, digitalization, and computational power have created an opportunity to build complex integrated hydrological models with reduced uncertainty.

Previous studies have shown that the urban subsurface is highly complex, with preferential flow paths due to anthropogenic fill material and flow barriers due to subsurface buildings (Salvadore et al., 2015; Hibbs and Sharp, 2012; Lerner, 2002, 1990), but few studies have documented the effects of the anthropogenic layer on groundwater at city scale. Moreover, even fewer studies use integrated modeling considering both surface and groundwater flows and their interaction with leaking water pipes and sewers.

This study investigates the impact of anthropogenic urban geology and spatial discretization on the simulation of shallow groundwater levels and flow paths at the city scale using an integrated hydrological model based on the MIKE SHE code. The Danish city of Odense is used as the case study for the model experiment. The objectives are (1) to develop two models of the urban geology, (2) to set up a hydrological model that integrates the surface-groundwater processes, as well as interactions with the sewer system, (3) to analyze the effect of the geological models on the simulation of groundwater levels

and flow paths, and (4) to analyze the effect of spatial discretization on the simulations.

## 2 Site description

The city of Odense is located near the coast in the south-central part of Denmark (Figure 1). The city dates back to 1580 and has mainly expanded since the beginning of the 19th century (Laursen and Linderberg, 2017). Figure 1 shows the areal extent of the model domain, which covers 9.8 km² and is an urban area that was developed in the 1960s-70s (Mielby and
Sandersen, 2017). Within the model domain, the terrain slopes from the west towards the northeast, with elevations from 45 to 5 meters above sea level (m.a.s.l.). There is one creek in the model named Hedebækken, see Figure 1a, which intersects the model boundary in the west and flows into the Odense River towards the east. Odense River makes up the eastern boundary of the model domain. Previous bog and marshland in the area have been drained and to a large degree urbanized since the 1950s (Laursen and Linderberg, 2017).

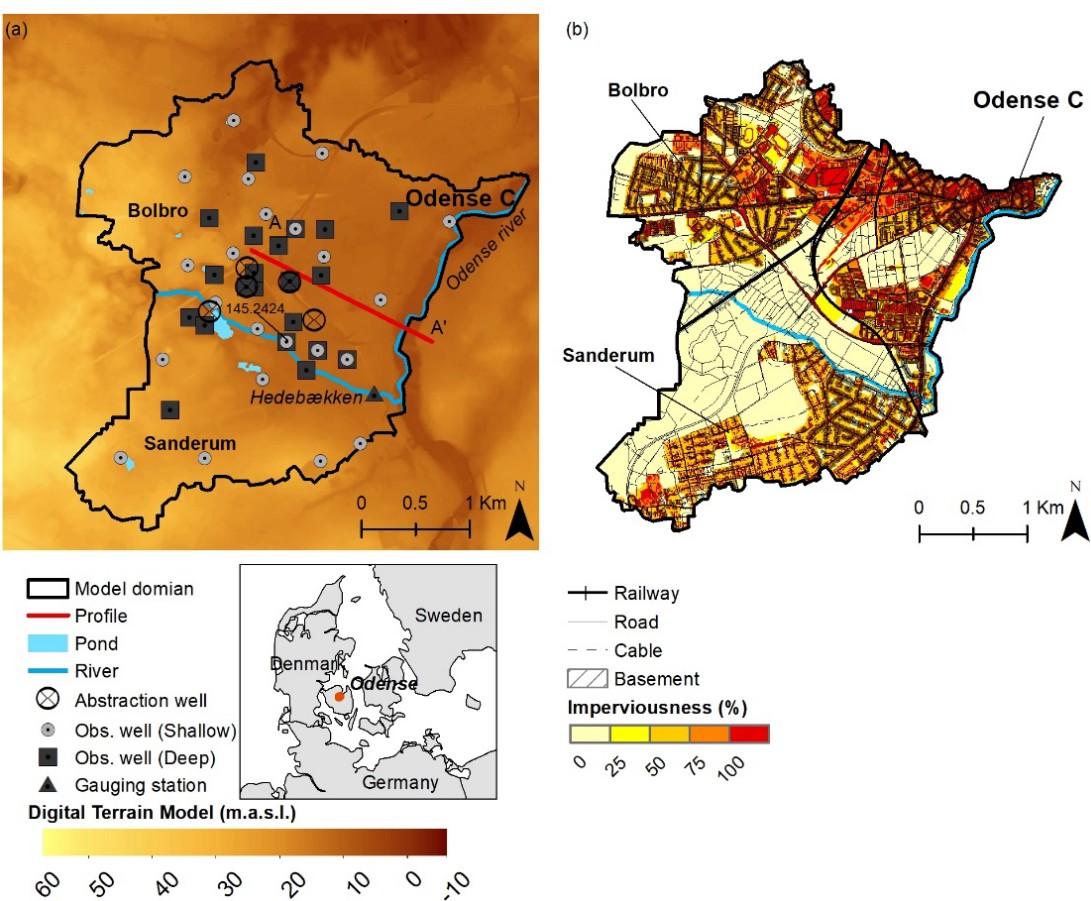


Figure 1. Maps of the model domain, a part of the city of Odense, Denmark. Panel (a) displays a digital terrain model from the Danish National Agency for Data Supply and Infrastructure (2019) with locations of observation points and abstraction wells provided by Vandcenter Syd A/S (2021). Panel (b) depicts the imperviousness of the land cover retrieved from Levin et al. (2012) and modified with data from Vandcenter Syd A/S (2019a) and subsurface infrastructure provided by Vandcenter Syd A/S (2019b)
and the Danish Geodata Agency (2019).

Figure 1b shows the imperviousness of the land cover and the location of infrastructure and basements. Approximately 50 % of the total land cover in the model domain is impermeable or semi-permeable such as buildings, asphalt, and concrete pavement, which have an imperviousness above 75 %. Buildings and pavements are normally considered 100% impervious. Yet, the map contains areas where the imperviousness is 75%. This can be areas where a little area with vegetation is placed next to a building or a road. This high percentage of imperviousness is dominant in the northern and eastern parts of the domain, see Figure 1b. The remaining land cover is unpaved, consisting of private and public green areas, minor deciduous forests, small ponds, and the two creeks. The distribution of the different land covers and the degree of imperviousness represent a typical urban area for larger Danish cities (population >100,000).

The length of the sewer pipe system within the model domain is 125 km and the majority of the pipes were established between 1960-1990 (Vandcenter Syd A/S, 2019b). The sewer system collects both stormwater runoff and wastewater; only in a few places in Odense, the two are separated. The drainage water generated in areas with separated systems is routed to the local rivers (Odense Kommune, 2011).

The local average precipitation is 761 mm year$^{-1}$, with a minimum of 41 mm in May and a maximum of 81 mm in October. The average reference evapotranspiration is 629 mm year$^{-1}$. The highest temperatures are in August with a daily average of 17.2 °C and the lowest in January with a daily average of 1.7 °C (DMI, 2021). Since evapotranspiration is low in the winter period (October-March), most of the groundwater recharge occurs in this period.

The landscape is a low-lying moraine landscape shaped by glaciations; in the northeastern part of the model domain, the landscape changes into a late-glacial plain (Mielby and Sandersen, 2017; Jakobsen and Tougaard, 2018). The predominant Quaternary deposits consist of glacial clay till - a mixture dominated by clay with contents of sand, gravel, and stones - and diluvial sand and clay deposited by glacial meltwater. The Quaternary deposits reach down to approximately -50 m.a.s.l. and locally even below -100 m.a.s.l., as seen in Figure 2.

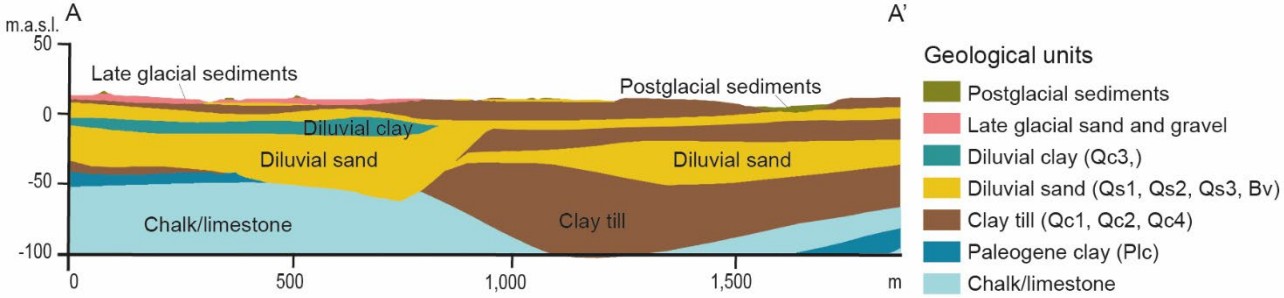

**Figure 2. Geological cross-section of profile A-A'. See Figure 1 for the location of the profile.**

At the terrain, the moraine landscape is dominated by clay till, while the late glacial plain is dominated by late-glacial sand and gravel, transported by meltwater from the last ice sheet (Weichselian). In the low-lying areas along the streams, postglacial deposits such as peat or gyttja dominate the shallow geology (Sandersen and Kallesøe, 2017; Jakobsen et al., 2022). In the urban area, the material composition of the first 1-5 meters can vary within short distances depending on anthropogenic activity. Both the shallow and deeper subsurface beneath the city are less well documented, compared to the

open land area. Below the Quaternary deposits follows a layer of impermeable Paleogene clay, which is overlaying Danian

limestone and Cretaceous chalk (Sandersen and Kallesøe, 2017).

The oldest Quaternary layers have locally been eroded by subglacial meltwater streams, which created buried valleys that in some areas reach down into the Paleocene clay and the limestone/chalk beneath (Sandersen and Jørgensen, 2016). Two buried valleys have been documented west and southeast of Odense. The oldest valley has a NE-SW orientation and the younger buried valley has an E-W orientation. Previous studies indicate that the two valleys unite just outside the Bolbro

district, see Figure 1, with a younger regional sand layer and it has been suggested that the E-W oriented valley continues east beneath the City of Odense (Sandersen and Kallesøe, 2017). The sandy infill of the two buried valleys and an overlaying regional sand layer are the primary aquifers used for water abstraction in Odense. Local sand and gravel deposits of the late glacial plains and the anthropogenic fill material form secondary shallow aquifers (Vandcenter Syd A/S, 2021). The shallow groundwater in the city of Odense threatens buildings and infrastructure when groundwater levels are high

which typically occurs in late winter and early spring. Since 2006 approximately 1-2 mill. $m^3$ $year^{-1}$ is pumped from five abstraction wells for the local water supply. These are located in the center of the model domain, outside the city center, see Figure 1. These wells extract water from the buried valleys and the regional sand layer.

## 3 Materials and methods

The effect of the geological configuration was analyzed by applying three geological models V0, V1, and V2 in otherwise

identical hydrological models. The effect of spatial discretization was tested by using a coarse discretization relative to the urban subsurface infrastructure, with a horizontal grid discretization of 50 m, and a finer discretization close to the scale of the urban subsurface infrastructure, e.g., roads and trenches, with a horizontal grid discretization of 10 m. For both the geological and hydrological modeling tools a discretization in the order of 1-10 m becomes computationally challenging when the size of the model is large, resulting in e.g., millions of grid cells. For the hydrological modeling, multiple grid sizes

were initially tested. The two grid sizes of 50 and 10 m were chosen based on a tradeoff between computation time and the number of grid cells for the model size and retaining geological detail. The effect of the geological configuration and spatial discretization was analyzed based on the hydrological models' simulation of high-water levels, the 95th percentile, and particle tracking.

### 3.1 Geological models

The geological model V0 was an existing hydrostratigraphical layer model developed for the larger Odense City area by Sandersen and Kallesøe (2017). The geological models V1 and V2 were developed as part of this study. An overview of the geological models and their differences is presented in Table 1.

**Table 1. Overview of models and their differences in discretization and geological model type**

| Hydrological model name | Horizontal discretization of the hydrological model (m) | Geological model name | Geological model type | Discretization of the geological model |
|---|---|---|---|---|
| V0_50 | 50 | V0 | Layer model | Layers and lenses with varying thickness |
| V0_10 | 10 | | | |
| V1_50 | 50 | V1 | Combined voxel and layer model with urban infrastructure | 5x5x1 m in the voxel model (the top 15 mbgl). Layers and lenses with varying thicknesses |
| V1_10 | 10 | | | |
| V2_50 | 50 | V2 | Combined voxel and layer model with urban infrastructure and soil material | 5x5x1 m in the voxel model (the top 15 mbgl). Layers and lenses with varying thicknesses. |
| V2_10 | 10 | | | |


The V0 model represents the Quaternary deposits and the pre-quaternary deposits of Paleogene clay and limestone/chalk down to -150 m.a.s.l. The interpretation of the upper geological layers did not include data on fill and anthropogenic materials. The geological models V1 and V2 represent the shallow urban geology down to 10 m below the terrain, in voxels, while below 10 m depth, the models retained the deeper stratigraphical layers from V0.

The method used for modeling the shallow urban geology in V1 and V2 follows the procedure presented by Mielby and Sandersen (2017). The method involves the following steps: (1) convert the upper layers of the geological layer model corresponding to the extent of the anthropogenic layer to voxels for a designated spatial discretization, and (2) assign a sand/clay fraction to each voxel corresponding to the materials' lithology, (3) implement anthropogenic data such as basements, pipelines and utility trenches in the voxel model and assign a sand/clay fraction to each data type, (4) for voxels

where anthropogenic data occur, compute the representative voxel value according to the volume.

The voxel models for V1 and V2 were constructed with a grid size of 5x5x1 m using a 3D floating-point grid in the geological modeling software GeoScene3D (www.i-gis.dk). The vertical extent of the voxel models was from the terrain to 10 meters below the lowest elevation point. The defined grid size was a trade-off between being able to represent the anthropogenic structures in the uppermost subsurface and computational constraints. The choice of discretization for the

voxel models with urban geology was guided by the experience from the study of Mielby & Henriksen (2020) and Mielby & Sandersen (2017) and chosen to be 5x5x1 m to be able to represent the subsurface infrastructure trenches which are typically 1-3 m wide and 1-2 m in depth for this study area, while the Road trenches are around 10 m wide +/- and therefore 5 m is a good intermediate size. The horizontal discretization was thus larger than the dimensions of the trenches. A smaller discretization for a model at the city scale would have been computationally expensive in the hydrological model. Mielby &

Sandersen (2017) argued that the discretization of the geological and hydrological model must meet the required detail, yet not exceed the computational capabilities. The two voxel models each had 22 million voxel grids.

    In model V1, data on urban subsurface infrastructure; utility trenches, pipes, roads and roadbeds, railway tracks, basements, and foundation materials were incorporated, based on the model procedure described above. The V2 voxel model included the same data on urban infrastructure as in V1 and additional data on soil material in the top 5 meters. The different data and

sources utilized for the geological models are presented in the supplementary material (Table S1). The additional data on soil material in V2 is a soil map (Jacobsen et al. 2022) and soil descriptions from shallow geotechnical boreholes (GEUS, 2019). The soil map by Jacobsen et al. (2022) is in 1:25000 resolution and is based on samples of soils every 200 m at 1 m depth. The soil descriptions from shallow geotechnical boreholes were derived by looking through non-digitalized documents in the Danish National well database. The modeling procedure for the V2 model of the shallow geology started with the input from

V0, followed by overwriting the upper first meter with soil data and modeling the fill layers based on descriptions from shallow geotechnical boreholes. A zone around the boreholes with a radius of 12 m was applied, where the fill material was assumed to be present. Finally, the anthropogenic infrastructure data also used in V1 was added.

    The sand/clay fractions in the voxels represent the expected content of sand and clay in the lithology of the geological layer or the equivalent properties for the subsurface infrastructure. The fraction values vary between 0 and 1 corresponding to

respectively, 100 % clay and 100 % sand.

**Table 2. Sand/clay fractions for the different geological units and infrastructures used in the voxel models**

| Type | Unit | Sand/clay fraction |
|---|---|---|
| Geological | Postglacial sediments | 0.5 |
| | Peat | 0.5 |
| | Gyttja | 0.3 |
| | Sand and gravel from late-glacial | 1 |
| | Clay from late-glacial, glacial lakes | 0 |
| | Freshwater clay, saltwater clay | 0.8 |
| | Freshwater sand, saltwater sand | 0.8 |
| | Diluvial silt | 0.4 |
| | Diluvial clay (Qc) | 0 |
| | Diluvial sand, diluvial gravel (Qs) | 1 |
| | Clay till (Qc) | 0.7 |
| Infrastructure | Trenches (mains and other pipelines) | 1 |
| | Roads (Base/embankments) | 1 |
| | Railroads (Base) | 1 |
| | Basements and foundations | 0 |
| | Fill | 1 |

    Assumptions of the sand/clay fractions were necessary since the description of the fill materials is often poor and often refers

to standards. The type of infill material used in construction depends on age as well as the type of infrastructure (Sandersen et al., 2015). The characteristics of the infill materials have not been recorded and the infill may change over very short

distances even for structures from the same period. The sand/clay fractions for the infrastructure classes, in Table 2, thus represent very diverse and inhomogeneous infills.

Each voxel was assigned a representative sand/clay fraction based on volume averaging of the sand/clay fractions representing the lithologies and the infrastructures occurring in the voxel, see Figure 3.

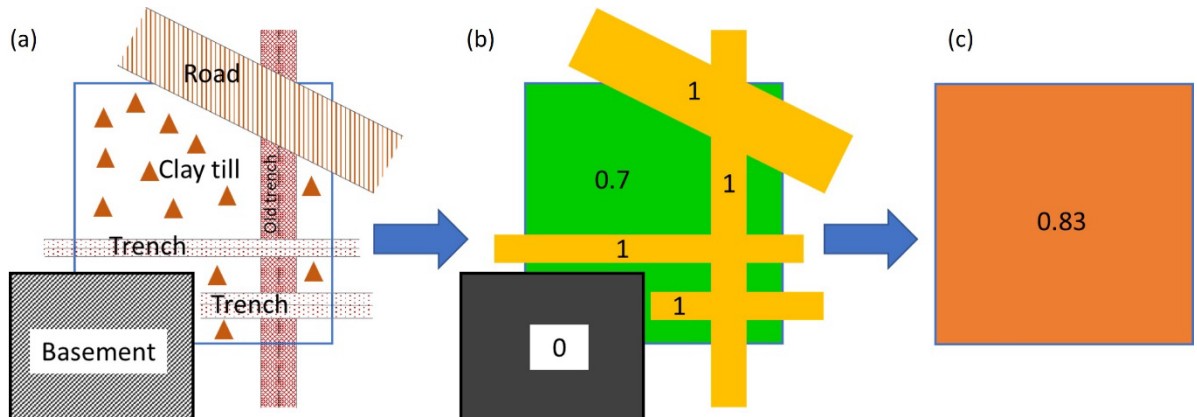

**Figure 3. Workflow for setting the value of a voxel: (a) Voxel with clay till and anthropogenic subsurface infrastructures, (b) The corresponding sand/clay fractions, and (c) the resulting representative sand/clay fraction for the voxel, which is based on average according to volume.**

The location of the roads and the pipes were retrieved from the national road directory and the pipe owners as GIS files, see table S1 for data sources. The extent of the excavations and trenches was based on national standards for profiles of road design and pipe trenches, see table S1 for sources. It was assumed that the design of the roads, railways, and trenches followed these standards. Data on infill materials and characteristics were estimated from borehole data in combination with information on age and knowledge of past and current standards for the size of trenches and the fill material in the excavations (Mielby and Sandersen, 2017). Data on subsurface electrical installations and other minor service lines were not considered as their spatial extent was negligible to the chosen model discretization.

### 3.2 Hydrological models

The study includes six hydrological models. The differences between the models were, as presented in Table 1, the application of two different horizontal discretizations and the three geological models used for the simulation of the subsurface processes. The hydrological models were based on the MIKE SHE code (Abbott et al., 1986 a,b). THE MIKE SHE code was chosen because of its ability to integrate the surface and subsurface processes dynamically, as well as its ability to include both overland and sewer drainage. Moreover, with this model code, the properties of the subsurface and the computational layers can be spatially distributed in both the horizontal and vertical planes. Other integrated hydrological models such as PARFLOW.CLM, MODFLOW 6, and HydroGeoSphere offer similar capabilities.

Figure 4 illustrates the common setup of the six hydrological models. The model components were overland flow, unsaturated zone flow, and saturated zone flow. The MIKE SHE models were coupled to a MIKE HYDRO model (DHI, 2017) for the simulation of groundwater seepage to the river bed and river discharge in the two creeks within the model domain.

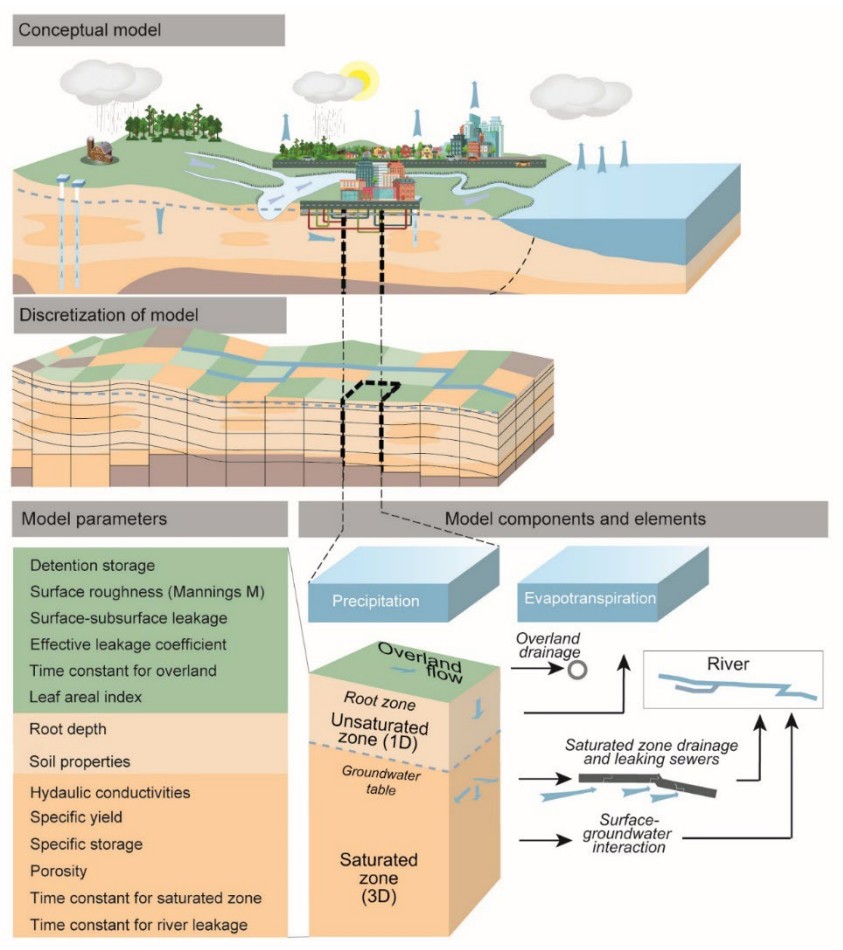


**Figure 4. Illustration of the model setup for the hydrological models**

The overland flow was described by a finite difference approximation of the 2D Saint Venant equations for diffusive flow. The unsaturated zone flow was described by a simplified two-layer water balance approach assuming vertical flow and a conceptual formulation for actual evapotranspiration. This approach is primarily applicable to areas where the groundwater

table is shallow and the actual evaporation rate is close to the potential rate (Butts and Graham, 2005), which is the case for the study site. The saturated zone flow was described by the governing equation for 3D saturated flow based on Darcy's law. Subsurface drainage was included as a sink term and depended on the groundwater level, depth of the drains, and a time constant. Detailed descriptions of the components can be found in DHI (2017, 2020).

The computational time steps were automatically controlled to secure accurate water balances. The maximum time steps
were: 0.5 hours for overland flow, 6 hours for unsaturated flow, and 12 hours for groundwater flow. The MIKE HYDRO
models used the kinematic wave approximation with a fixed time step of 10 minutes.

### 3.2.2 Model input and parameterization

The models were driven by daily values of precipitation, temperature, and reference evapotranspiration, which were
retrieved as daily averaged gridded data (10 x 10 km) from the Danish Metrological institute (DMI, 2021). A digital terrain
model (DTM) in 10 m discretization (Danish National Agency for Data Supply and Infrastructure, 2019) was used as the
surface elevation and a land-use map in 10 m raster discretization from Levin et al. (2017) was used as a basis for the
vegetation characterization in the models. This land-use map contains 36 land-use classes which were reclassified into 6
classes: buildings, grass, deciduous forest, coniferous forest, agriculture, and water bodies. Buildings and grass are the
dominant classes in the domain. For each land-use class, a monthly variation of leaf area index and root depth was defined.
The initial parameter values were retrieved from the National hydrological model of Denmark (DK-model) (Stisen et al.,
2019).

Data on imperviousness was derived from land cover data (Levin et al., 2012) in 10 x 10 m discretization and updated with
the local water supply's map of imperviousness (Vandcenter Syd A/S, 2019a). In the resulting imperviousness map see
Figure 1a, the data were grouped in quartiles and used as the paved area fraction for each grid cell in the models. The paved
area fraction was used to define the overland drain runoff coefficient of ponded water and the classification of detention
storage. Moreover, it was used as a linear scaling fraction for the surface-subsurface leakage coefficient. The surface-
subsurface leakage coefficient reduces the infiltration from the surface to the subsurface at paved surfaces as well as the
seepage from the subsurface to the surface. In the model, it is applied to the areas where the paved area fraction is above 0.5
and it was given the value $6 \times 10^{-7}$ $s^{-1}$ in all cells and then scaled by the paved area fraction. The scaling of the surface-
subsurface leakage coefficient by the paved area fraction is referred to as the effective leakage coefficient (DHI, 2020).

Overland flow occurs when the detention storage capacity is exceeded, representing the depth of water retained in local
depressions due to heterogeneities/roughness of the land surface. The detention storage was set to 1 mm for cells with a
paved area fraction $\geq$ 0.75 assuming a very smooth surface, while the remaining grid cells were given a value of 4 mm.
Overland flow is controlled by the surface roughness described by the Manning M which was set to 10 $m^{1/3}$ * $s^{-1}$ for the
entire domain.

Drainage of overland water was assumed to occur from paved road networks and larger sealed surfaces. Cells with a paved
area fraction larger than 0.5 were considered to be part of the overland drainage network, whereas cells with a lower paved
area fraction were assumed to be green areas and unattached to the overland drainage system. The time constant for the
overland drains was set to 0.001 $s^{-1}$. This is a relatively high leakage coefficient, which in most cases will drain all available
ponded water from the cell.

The subsurface component of the hydrological models had 14 computational layers. The thickness of the computational layers and their outer boundary conditions is specified in Table 3, and even though the geological models were altered the computational layer settings were kept constant. The hydraulic properties of the subsurface system were assigned to the geological units within the computational layers. This made it possible to assign individual parameter values to the
stratigraphic layers, lenses, and the sand/clay fraction classes in the V1 and V2 models.

The boundaries of the model domain follow natural boundaries in the upper 8 meters of the model. The boundary conditions for the conductive layers below 8 meters were specified using a nested modeling approach, where a regional model (Kidmose and Sonnenborg, 2018) was used to calculate head boundary conditions for this city-scale domain, Table 3.

**Table 3: Conditions for the computational layers in the hydrological models**

| Computational layer no. | Layer thickness (m.b.g.l.) | Hydrogeology in the V0 model (abbreviation for the hydrostratigraphical layer) | Outer boundary condition* |
|---|---|---|---|
| 1 | 0-1 | | Zero flux |
| 2 | 1-2 | | Zero flux |
| 3 | 2-4 | Postglacial soils, late glacial sand, and gravel | Zero flux |
| 4 | 4-6 | | Zero flux |
| 5 | 6-8 | Clay till (Qc1) | Zero flux |
| 6 | 8-10 | Diluvial sand and gravel (Qs1) | Specified head |
| 7 | Distributed | Clay till (Qc2) | Zero flux |
| 8 | Distributed | Diluvial sand and gravel (Qs2) | Specified head |
| 9 | Distributed | Diluvial clay (Qc3) | Zero flux |
| 10 | Distributed | Diluvial sand and gravel (bv) | Specified head |
| 11 | Distributed | Clay till (Qc4) | Zero flux |
| 12 | Distributed | Diluvial sand and gravel (Qs3) | Specified head |
| 13 | Distributed | Paleogene clay (Plc) | Zero flux |
| 14 | Distributed | Limestone/Chalk | Specified head |

*A zero flow condition was assumed for all layers along the eastern boundary of the domain since this part follows the Odense River.

Since the objective was to analyze the impact of the shallow urban geology, the vertical discretization of the computational layers was set to 1 meter at the top and increases with depth until layer six, 10 m below the terrain. From layer seven and down, the computational layer thickness follows the extent of regional layers from the regional model. The small layer
thickness at the top of the model was specified to resolve the variation of the anthropogenic geology in the geological models V1 and V2. The layer thickness was set to increase with depth because the anthropogenic effect is expected to decrease with depth and a coarser computational grid reduces the computational burden.

Saturated zone drainage was represented in the models at distributed depths based on three assumptions: (1) the sewer system in the impervious part of the urban model was leaky and the sewer pipes act as saturated zone drains if the groundwater level reaches the level of a sewer pipe, (2) buildings with basements all have perimeter drains installed at 3 m b.g.l. (below ground level) except for basements at two locations, where the basement reaches down to respectively 4 and 10 m b.g.l., and (3) in green and forested areas as well as open landscapes drain depth was assumed to be 1 m b.g.l.

The location and depth of the sewer system were provided by the local wastewater company (Vandcenter Syd A/S, 2019b) and data on buildings with basements were received from the municipality (Odense Kommune, 2019). In the model setup, the drainage of groundwater into the sewers and the drains was assumed to be the same for the entire domain. The water collected by the drains is discharged to either one of the two streams in the model, or out of the model if the area is not a part of a contributing area to the streams.

## 3.3 Calibration

The hydrological models were calibrated using the parameter estimation tool PEST (Doherty, 2016b, a) following the same procedure for all 6 models. In the calibration scenarios, the models were run for the period 2012-2020, where 2012-2014 was used as a warm-up and 2015-2020 as the calibration period.

Time series of observations of the hydraulic head from 50 wells and stream discharge from one station were used for calibration. The local water supply company Vandcenter Syd A/S (2021) provided 16 hydraulic head time series from both shallow and deep aquifers, which covered the entire calibration period, while the remaining 14 time series were from shallow wells that were established in 2019 as part of this study and therefore these time series only covers the year 2020 of the calibration. The discharge time series covers the entire calibration period and was derived from a Q/h rating curve of stream measurements undertaken in 2019 and 2020 from the Hedebækken river as part of this study. The location of the observation wells and stream gauge is shown in Figure 1. Several of the observation wells had 2-4 well screens for observation of deep and shallow aquifers. Half of the well screens were assumed to be placed in shallow aquifers due to their shallow depth and the lithological characterization from the boreholes.

The parameters selected for calibration (free parameters) and the ones tied to the calibration parameters were based on a sensitivity analysis, where all model parameters were analyzed. The sensitivity analysis was based on composite sensitivities as described in Doherty (2015) and conducted for all 96 model parameters. Based on the analyses across all models a set of free parameters was selected subject to calibration and another set of parameters to be tied to the free parameters. One parameter set was selected for the V0 model and one parameter set was selected for the V1 and V2 models, see Table 4.

It was chosen to tie the vertical and horizontal conductivities and the sand/clay fraction classes to maintain their linear relationship with increasing sand content. One can argue that the sand/clay fraction was calibrated as one class. For parameters with little sensitivity and not subject to calibration, their values were instead estimated from past model experiences.

**Table 4. Parameter sets for calibration, h stands for horizontal, v stands for vertical, and the S10-S100 stands for the sand/clay fraction classes, where e.g., S100 is the sand/clay fraction 0.9-1.**

| Calibration parameter set for V0 models | | | Calibration parameter set for V1 and V2 models | | |
|---|---|---|---|---|---|
| Description of free parameter | Abbreviation of free parameter | Tied parameter(s) | Description of free parameter | Abbreviation of free parameter | Tied parameter(s) |
| Horizontal hydraulic conductivity for Quaternary buried valley | $K_{bv2,h}$ | $K_{bv2,v}$ | Horizontal hydraulic conductivity for Quaternary buried valley | $K_{bv2,h}$ | $K_{bv2,v}$ |
| Horizontal hydraulic conductivity for Quaternary sand 2 | $K_{Qs2,h}$ | $K_{Qs2,v}$ , $K_{Qs3,h}$ , $K_{Qs3,v}$ | Horizontal hydraulic conductivity for Quaternary sand 2 | $K_{Qs2,h}$ | $K_{Qs2,v}$ , $K_{Qs3,h}$ , $K_{Qs3,v}$ |
| Vertical hydraulic conductivity for Quaternary clay | $K_{Qc,v}$ | $K_{Qc,h}$ | Vertical hydraulic conductivity for Quaternary clay | $K_{Qc,v}$ | $K_{Qc,h}$ , $K_{S20,h}$ , $K_{S20,v}$ $K_{S30,h}$ , $K_{S30,v}$ $K_{S40,h}$ , $K_{S40,v}$ |
| Saturated zone drain time constant | $C_{dr}$ | | Saturated zone drain time constant | $C_{dr}$ | |
| Root depth of permanent grass | $D_{r,grass}$ | $D_{r,decid}$ | Root depth of permanent grass | $D_{r,grass}$ | $D_{r,decid}$ |
| Leakage coefficient, Rivers | $C_{riv}$ | | Leakage coefficient, Rivers | $C_{riv}$ | |
| Horizontal hydraulic conductivity for Clay till | $K_{ct,h}$ | $K_{ct,v}$ | Horizontal hydraulic conductivity for sand/clay fraction 0.6-0.7 | $K_{S70,h}$ | $K_{S100,h}$ , $K_{S100,v}$ $K_{S90,h}$ , $K_{S90,v}$ $K_{S80,h}$ , $K_{S80,v}$ $K_{S70,v}$ , $K_{S60,h}$ , $K_{S60,v}$ $K_{S50,h}$ , $K_{S50,v}$ |
| Specific storage for Quaternary buried valley | $S_{s,bv2}$ | | | | |

For calibration, Tikhonov regularization based on preferred values in combination with Singular Value Decomposition (SVD) was used (Doherty, 2015). To account for regularization a Tikhonov regularization term was included in the objective function, which was minimized as a compromise between minimizing the measurement objective function and the

regularization term representing the deviation from the initially preferred parameter values (Doherty, 2015). The total objective function, $\Phi_t$, is given in Eq. (1).

$$\Phi_t \ = \ \Phi_m + \ \mu^2\Phi_r \tag{1}$$

where $\Phi_m$ is the measurement objective function given by Eq. (2), $\Phi_r$ is the regularization objective function and $\mu^2$ is the

global regularization weight factor.

The measurement objective function was defined as the sum of squared weighted residuals of hydraulic heads (h), head amplitude (ampl), and stream discharge (d):

$$\Phi_m = \alpha_h * \sum_{i=1}^{h}(\omega_{h,i}(h_{obs,i} - h_{sim,i}))^2 + \alpha_{ampl} * \sum_{j=1}^{ampl}(\omega_{ampl,j}(ampl_{obs,j} - ampl_{sim,j}))^2 + \alpha_d *$$

$$\sum_{k=1}^{d}(\omega_{d,k}(d_{obs,k} - d_{sim,k}))^2 \tag{2}$$

where α is the group weight. The group weight was assigned as follows: α_h=0.45, α_ampl=0.45, and α_d=0.10. ω_is the weight of i'th, j'th or k'th observation. As a standard, all observations had a weight of 1, yet after initial calibration runs some of the head and amplitude observations located near the west boundary were assigned a value of 0. For optimizing the parameters, more weight was given to hydraulic head observations than to discharge observation data. The following group weights were assigned in the objective function Eq. (2): head (0.45), head amplitude (0.45), and discharge (0.10).

The regularization objective function was defined by

$$\Phi_r = \sum_{i=1}^{p}(\mu_{p,i}^2(p_{ini,i} - p_{adj,i}))^2 \tag{3}$$

where p is the parameter-set, $p_{ini,i}$ is the preferred value for parameter i, $p_{adj,i}$ is the estimated value for parameter i, and $\mu_{p,i}^2$ is the differential regularization weight factor (Doherty, 2015).

The regularization weight factor $\mu^2$ was constrained by a target measurement objective function $\Phi_m^t$, which implies that the 355 model is considered calibrated when the measurement objective function is equal to or below $\Phi_m^t$. In addition, an acceptable measurement objective function $\Phi_m^a$ is defined and below which the calibration is accepted.

The selection of appropriate target and acceptable measurement objective functions $\Phi_m^t$ and $\Phi_m^a$ is subjective and must be assessed during the calibration process (Doherty, 2015). The target and acceptable objective measurement function were defined by first running the optimization without regularization and then subsequently specifying $\Phi_m^t$ to 5-10% higher than 360 the objective function obtained from the initial run. The acceptable measurement objective function $\Phi_m^a$ was specified 5% higher than the target measurement objective $\Phi_m^t$ as suggested by Doherty (2015). An overview of the hydrological models, their differences in geology and spatial discretization, and the calibration setup can be found in the supplementary material (Table S2).

The goodness of fit for the calibrated models was assessed based on the average error statistics and the spatial distribution of 365 the mean error (ME) of simulated heads. The parameter uncertainty was estimated based on linear statistics and using the measures of parameter identifiability, comparison of estimated parameters, and prediction uncertainty as suggested by Doherty (2016b).

### 3.4 Particle tracking

Particle simulations were carried out for all the calibrated models to compare the distributions of travel time and the 370 distribution of the particles at end sinks across the different model versions. Forward particle tracking was based on interpolated velocity fields, where the porosity was set to 0.2 for sandy units and 0.3 for clay-dominated units. The particle tracking was carried out for 200 years looping the dynamic flow fields for the calibration period 2015-2020. Particles were

released at the initial timestep uniformly in each cell from layers 1-6. They were thus released throughout the anthropogenic model layers from terrain and down to 10 m.b.g.l. To be consistent and have a comparable number of particles across the models with different grid cell sizes, only one particle was released in each cell in the 10 m models, while 25 particles were released in each cell in the 50 m models. A total of 625,300 particles were released in the 50 m models and 595,618 in the 10 m models. If particles were released above the water table, they were immobilized and not part of the analysis.

## 4 Results

### 4.1. Geological models

The A-A' cross-section from the initial geological layer model, V0, a voxel model of V0, and the two urban voxel models V1 and V2 are shown in Figure 5. The V0 voxel model was used as a step before the infrastructure data was added and converted to model V1, but the V0 voxel model was not used in the hydrological model. Yet, it was decided to compare the V0 voxel model with the V1 and V2 voxel models to make a quantitative comparison between the models, see Figure 6. The differences between the models mainly occur in the uppermost 1-3 m.b.g.l. where most of the anthropogenic structures primarily are located. From Figure 5, notably, models V1 and V2 have more sand/clay fraction classes embedded as opposed to the voxel model V0 where only 4 classes are present.

Overall, the V2 voxel model is not very different from the V1 voxel model. In the V2 model, the sandy material became sandier and the low-permeable material became less permeable, see Figure 5c-d. The geological structures deeper than 5 m below the surface remained the same in all three models, except at a few locations where additional borehole data suggested that a basement under a hospital extended deeper. Therefore, the two dominant voxel classes in the deeper layers, the sand/clay fractions 0.6-0.7 and 0.9-1.0, have approximately the same relative volume in all three models, 37% and 56%, respectively. Thus, the difference between models is only apparent in the remaining 7% of the total volume.

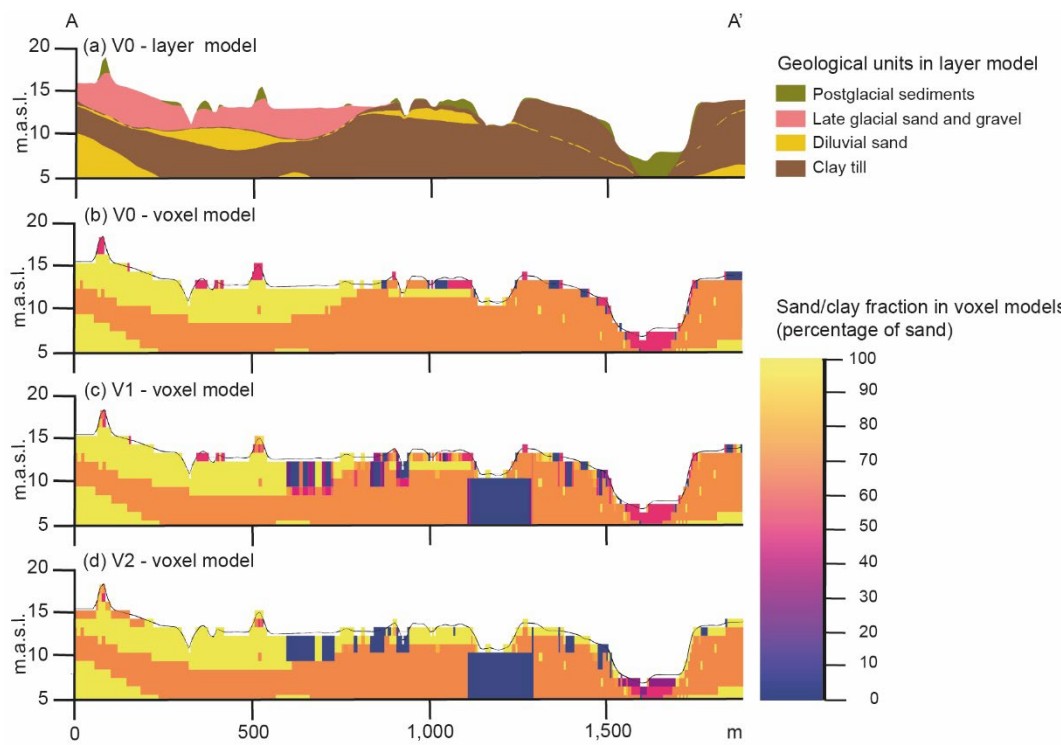

Figure 5. Profile A-A' from the geological models V0 – layer model (a), V0 – voxel model, (b) V1 – voxel model (c), and V2 – voxel model (d).

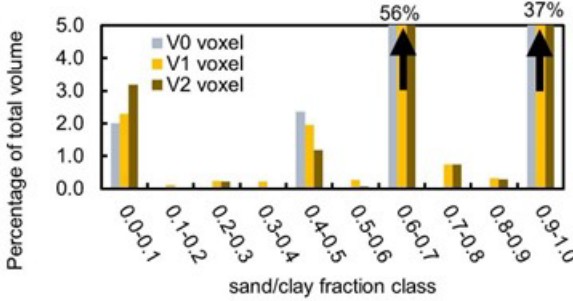

Figure 6. Percentage of the sand/clay fractions of the total volume of the geological voxel models V0, V1, and V2. Note that the sand/clay fractions 0.6-0.7 and 0.9-1.0, respectively take up about 37 % and 56 % in all models, and plot beyond the range of the y-axis.

The largest differences between the relative volumes of the voxel fractions are seen for the two classes 0.0-0.1 and 0.4-0.5. The relative volume of the most impermeable fraction class 0-0.1 increased slightly as the infrastructure data was added in V1, while it increased more than one-third between V0 and V2. On the contrary, the relative volume of fraction class 0.4-0.5 decreased as data on infrastructure and near-terrain geology were added in V1 and V2.

## 4.2 Calibration of hydrological models

Figure 7 depicts the three error statistics, i.e. the root-mean-square error of heads (RMSE_h ) and stream discharge (RMSE_Q), and residual of yearly groundwater head amplitude (ErrAmpl_h). The error statistics indicate an overall good performance for all models and a similar level of fit to the head and discharge observations. Nevertheless, RMSE_h for the V1_10 and V2_10 models was 20 cm lower compared to the other models while ErrAmpl_h was about the same and RMSE_Q slightly higher.

Analyzing RMSE_h of the shallow and deep well screens individually, RMSE_h was 1.0 m for the shallow well screens in the urban models and 1.2 m for the shallow well screens in the V0 models. On the other hand, RMSE_h was 1.7-1.8 m for the deep wells in the urban models, and 1.6 m and 1.8 m for the two models V0_50 and V0_10, respectively.

ErrAmpl_h varied between 0.2 m and 0.3 m across the different model versions. For V0_10 and V0_50 ErrAmpl_h was 0.2 m, while for all the V1 and V2 models the ErrAmpl_h was 0.3 m, see Figure 7. This appears as a reasonable accuracy since

most of the head time series has an annual amplitude of 0.5-1.0 m.

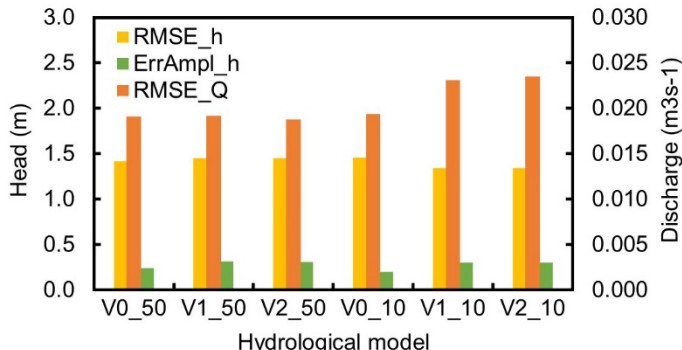

**Figure 7. A plot of simulated error statistics based on three metrics, i.e., root-mean-squared-error (RMSE) of heads (h), RMSE of discharge (Q), and error of yearly groundwater head amplitude (ErrAmpl_h).**

RMSE_Q was 0.019 $m^3s^{-1}$ for all models in 50 m discretization and V0_10, while V1_10 and V2_10 have slightly higher RMSE_Q values, see Figure 7. The calibration was found to be acceptable since the average discharge observation is 0.08 $m^3s^{-1}$, and the accuracy of the measurements is small because of an uncertain rating curve. Furthermore, discharge was only weighted by 10% in the objective function.

As an example of the model accuracy, Figure 8 shows the simulated and observed hydraulic heads for the six models for one of the shallow observation wells.

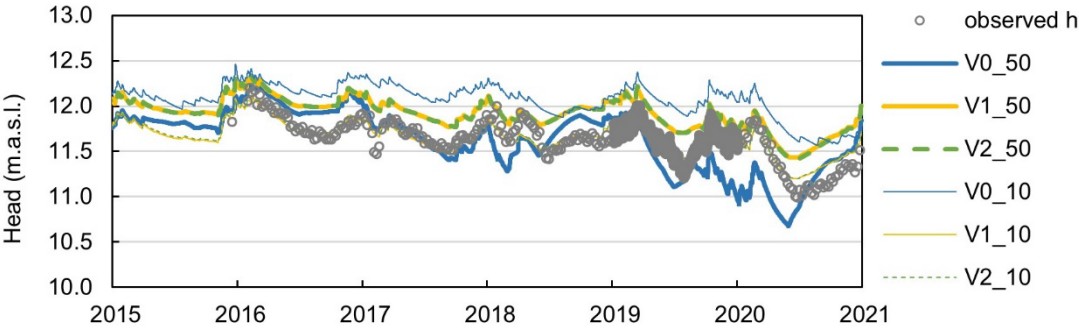

**Figure 8. Observed and simulated head for a shallow well screen (145.2424_3, see Figure 1)**

The spatial distribution of mean error (ME) for simulated heads is depicted in Figure 9 and generally shows similar tendencies. The ME is within +/- 1 m for most of the shallow well screens with a few outliers, and within +1 to -2 m for the deep screens. The best fit to the simulated heads was obtained in the central part of the model area. However, here more observation wells are available with time series that cover the whole calibration period. All models underestimate (orange and red colors) the groundwater level in the north-western part of the model area, while most of the models also have issues with underestimated groundwater levels close to the boundary in the southeastern part of the model area. The underestimation in the north-western part may be related to perched water tables not considered in the models and because of the scarcity of wells in this area.

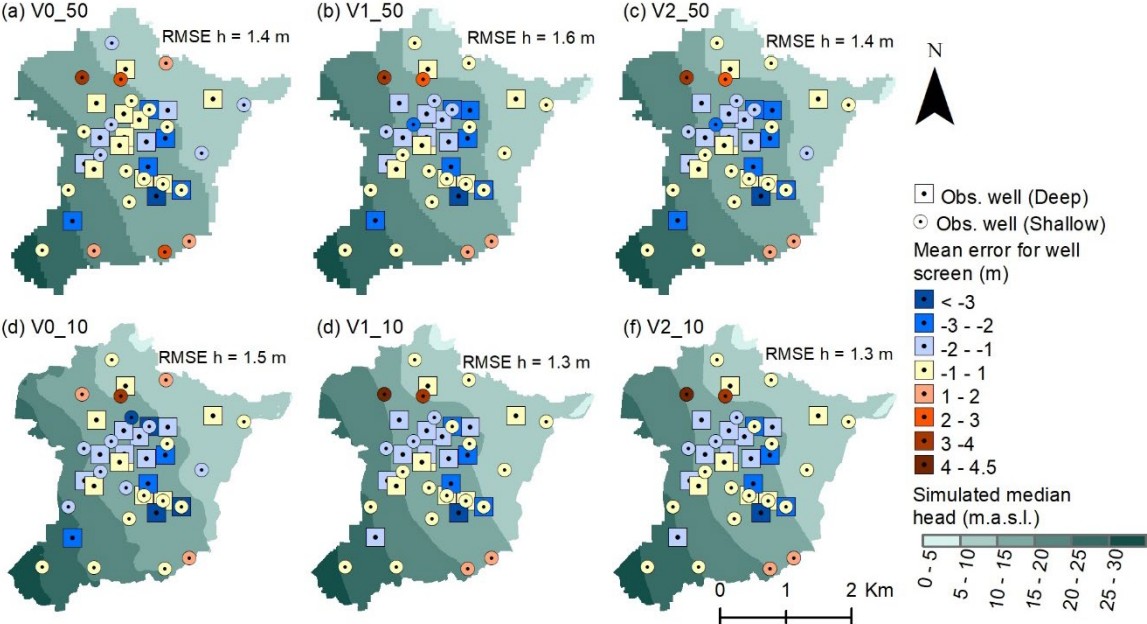

**Figure 9. Mean error of simulated hydraulic head, where the circle signature represents the shallow well screens and the square signature represents the deep well screens. The background map shows the median simulated head.**

The calibration setup ensured that the models achieved a measurement objective function $\Phi_m$ within the range of the target $\Phi_m^t$ and acceptable $\Phi_m^a$ objective function. Figure 10 shows the preferred initial parameter values (- symbols) together with the estimated parameter values with regularization (o-symbols) and without regularization (x-symbols). Many of the parameter estimates are different from their initial values when the models were calibrated without regularization while the regularization as expected made some of the parameters come closer to the preferred values. When regularization was used,

the parameters of hydraulic conductivities $K_{ct, h}$, $K_{Qs2,h}$, and $K_{Qc,v}$ in the V0_50 model show the largest difference from the initial parameters. In contrast, all parameter values estimated in the V0_10 model were close to the initial values. For V0_50 only, the discrepancy between the estimated vertical hydraulic conductivity of quaternary clay $K_{qc,v}$ and the initial value was larger when the models were calibrated with regularization.

The parameter estimates of the V1 and V2 models are generally close to the initial parameter values, see Figure 10. When

regularization was included the largest deviation from the initial values occurred for the hydraulic conductivity of the two sandy materials $K_{qbv2, h}$ and $K_{s70,h}$, with a factor of 0.8 and 0.6, respectively.

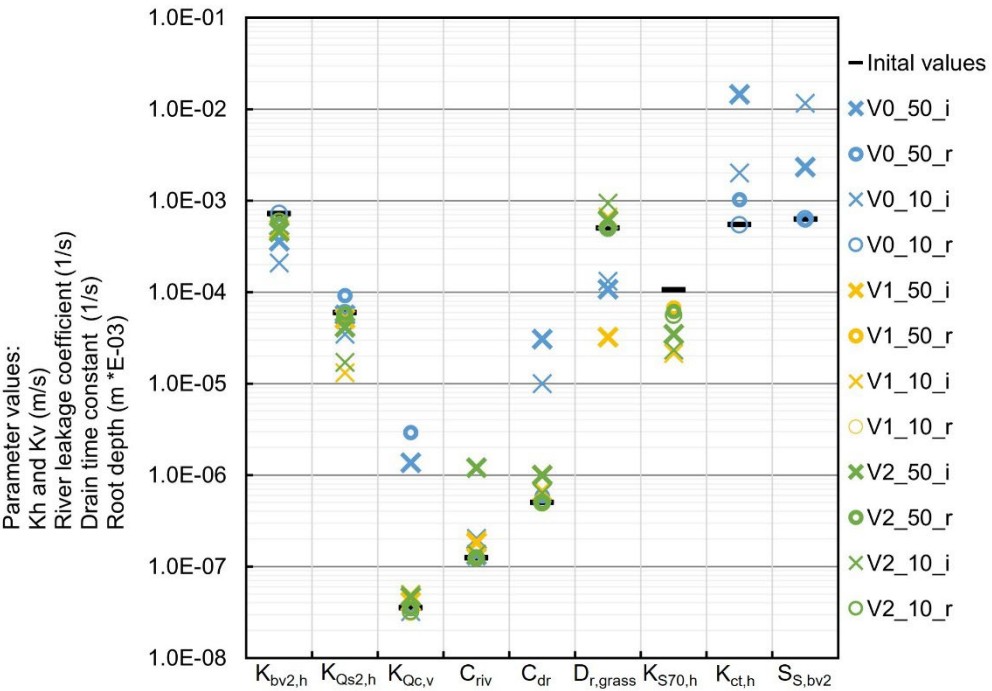

**Figure 10. Initial preferred parameter values and estimated values of free parameters for models calibrated without regularization (i) (x-symbol) and models calibrated with regularization (r) (o-symbol).**

The bar charts in Figure 11 illustrate the contribution of the singular values to the identifiability and indicate which parameters' relative error variance can be reduced by calibration (Doherty, 2015). Since the parameters' identifiability was not truncated all parameters have a value of one.

Red-yellow colors indicate that the parameter is informed by observation data and that the parameter value's relative error variance can be reduced through calibration. Blue-green colors, on the other hand, indicate that the parameter estimation is less informed by the observation data, hence the relative error variance cannot be reduced by calibration (Doherty, 2015).

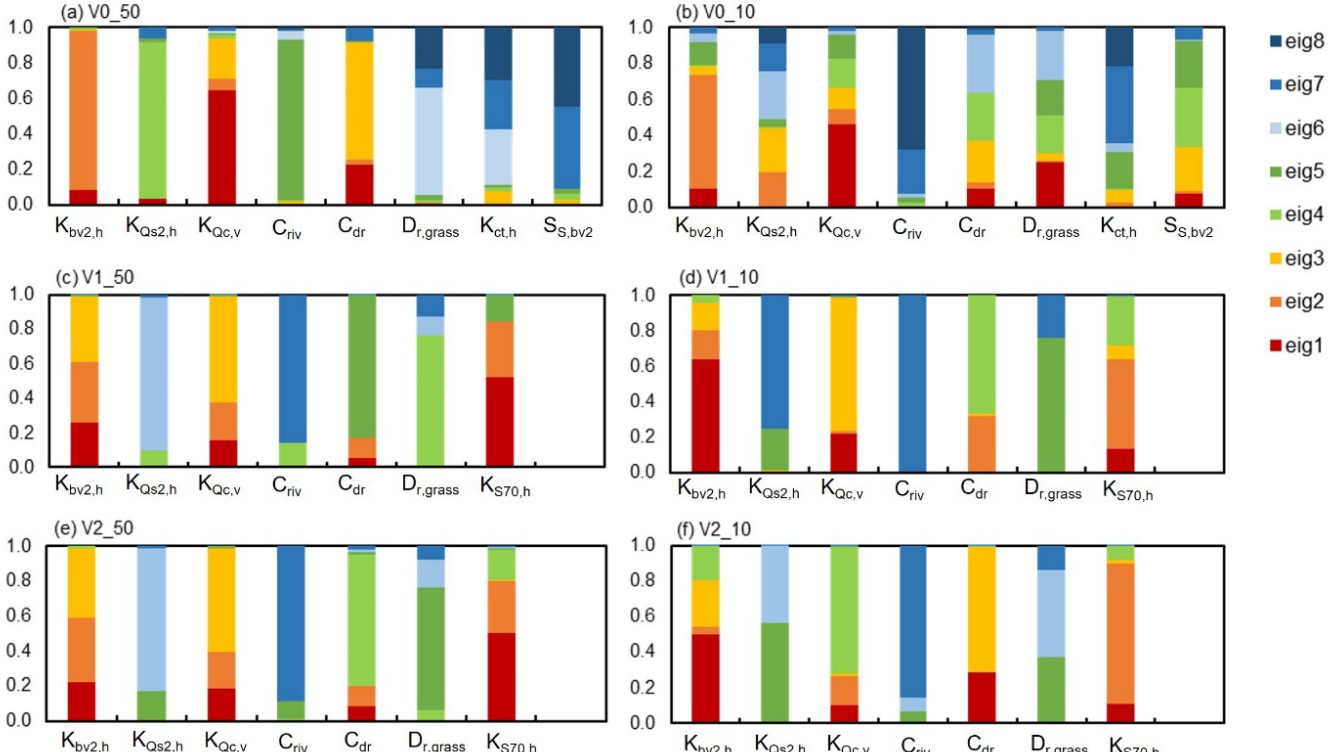

**Figure 11. Parameter identifiability without truncation of the solution space. The colors indicate the contribution made to the total identifiability by different eigen vectors in the solution space.**

Figure 11a-b depict the parameter identifiability for the models V0_50 and V0_10 and show that the hydraulic conductivities $K_{bv2,h}$ and $K_{Qc,v}$ are well informed by measurements since they are dominated by red-yellow colors. For V0_50 the parameters dominated by blue-green colors are $K_{ct,h}$ and $D_{r,grass}$, while for V0_10 it is $K_{ct,h}$ and $C_{riv}$. For the V1 and V2 models the parameters $K_{bv2,h}$, $K_{s70,h}$, and $K_{Qc,v}$ are mostly dominated by red-yellow colors, while $K_{Qs2,h}$, $C_{riv}$, and $D_{r,grass}$ are dominated by blue colors. Furthermore, for both models, the proportion of red color increases for $K_{bv2,h}$ when the model is run in 10 m discretization compared to the 50 m discretization. On the contrary, the proportion of red color decreases for parameter $K_{S70,h}$, when the model is in 10 m discretization compared to 50 m discretization.

The parameter values that deviate most from their preferred values are those that are dominated by red and orange colors in Figure 11 while parameters with more blue-green and more mixed color patterns obtain parameter values closer to their preferred value. This illustrates that the parameters which were less informed by the observations and less subject to a

reduction in error variance by calibration are the ones dominated by blue to green colors. The parameter values obtained by the regularized calibration were considered realistic and well-defined and were accepted for further analysis.

## 4.3 Simulation of high-water levels

The 95[th] percentile of the simulated water table depths for all six models is shown in Figure 12. All panels in Figure 12 show the same general pattern. 0-2 m water table depth in the lowlands and along the rivers, and above 10 m depth in the northwestern part and south part of the model domain. The 50 m models resulted in very similar results in Figure 12a-c, especially the models V1_50 and V2_50. Yet, the models V1_50 and V2_50 resulted in a slightly less depth to the water table in the areas with high imperviousness in the north and the center of the model domain, than the model V0_50.

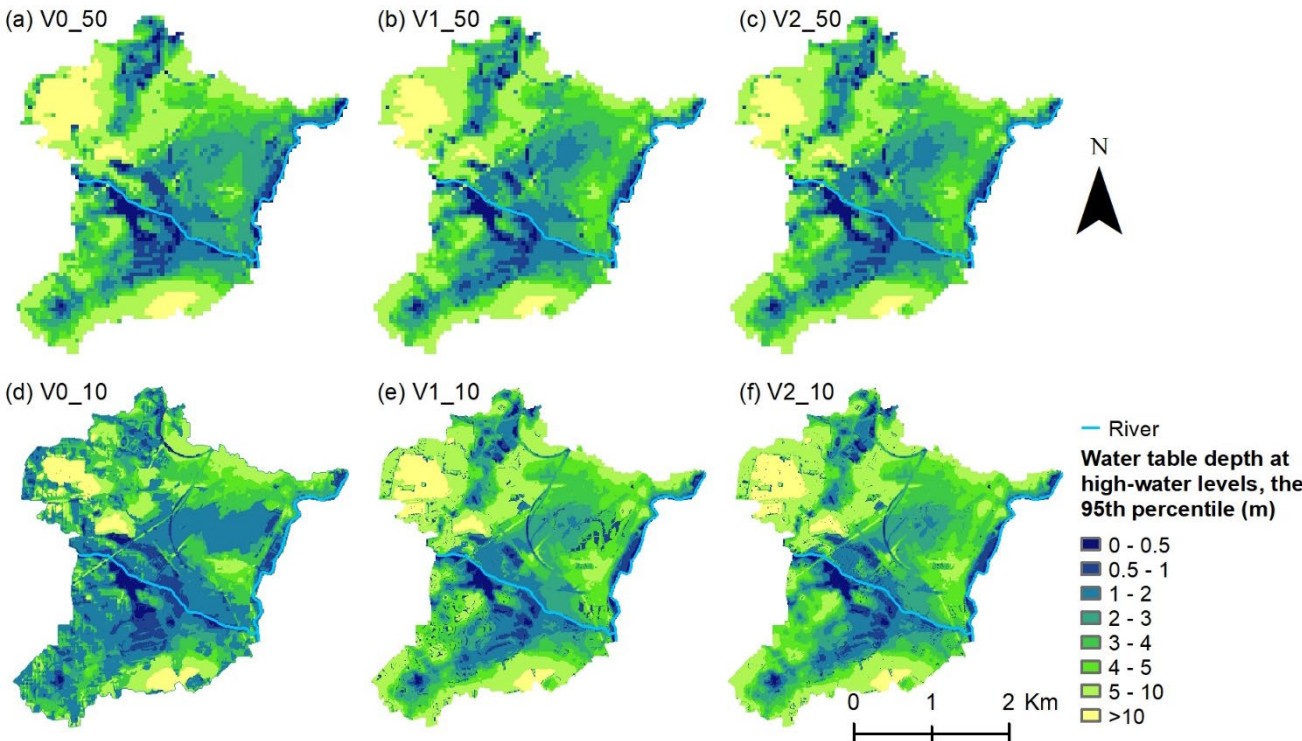

**Figure 12. The 95th percentile of the simulated water table depth**

The high-water simulation from the 10 m models V1_10 and V2_10 again showed very similar results, see Figure 12 d-f, with a small deviation from the 50 m models, yet the finer discretization allowed for more local variation. Meanwhile, the model V0_10m showed the largest area with the water table depth ranging from 0-2 m, see Figure 12d.

## 4.4 Particle tracking

In Table 5, the average, median, and 5th and 95th percentiles of travel times for particles reaching a sink are listed. Note that some particles were captured in the unsaturated zone and not tracked in the subsurface and as such represents one sink type. Other sinks include drainage to stream, drainage to the model boundary, and baseflow to rivers.

**Table 5. Statistics on simulated particle tracking for particles reaching sinks.**

|  | Average travel time (yrs) | Median travel time (yrs) | 5th percentile of travel time (yrs) [days] | 95th percentile of travel time (yrs) | Percentage of released particles registered in layer 8 (regional sand aquifer) (%) |
|---|---|---|---|---|---|
| V0_10m | 37.2 | 8.6 | 0.06 [21] | 158 | 25 |
| V1_10m | 7.3 | 2.4 | 0.06 [22] | 24 | 8 |
| V2_10m | 7.0 | 2.4 | 0.07 [25] | 21 | 9 |
| V0_50m | 37.0 | 22.7 | 0.10 [35] | 133 | 34 |
| V1_50m | 17.7 | 4.5 | 0.08 [31] | 106 | 12 |
| V2_50m | 17.6 | 4.6 | 0.09 [33] | 103 | 13 |

Tracking of particles to sinks resulted in a right-skewed distribution, with 5-10% of particles reaching a sink within 0.25 years and 25-60 % of the particles reaching a sink within 10 years depending on the model version. The 10 m models resulted in a smaller median travel time than the corresponding 50 m models and the urban models V1 and V2 had a smaller median travel time to sink than in the V0 models for both 10 m and 50 m discretizations. The average travel time for V0_10 and V0_50 were both 37 years while the median was estimated to be 22.7 years and 8.6 years, respectively. This indicates that the travel time distribution for V1_10 is more skewed towards longer travel times than the V0_50 model.

The number of particles reaching the deeper groundwater is much higher for models V0_10 and V0_50 than in the other models which had urban geology represented, see Table 5.

Figure 13 depicts the release location of particles that represent the 5[th] percentile of travel times to sinks. Again, the largest differences are seen between the V0 models and the two urban models V1 and V2. When the models are run in 50 m discretization the locations of the V1 and V2 models overlap (Figure 13b-c) while for 10 m discretization, the release locations show some differences (Figure 13e-f).

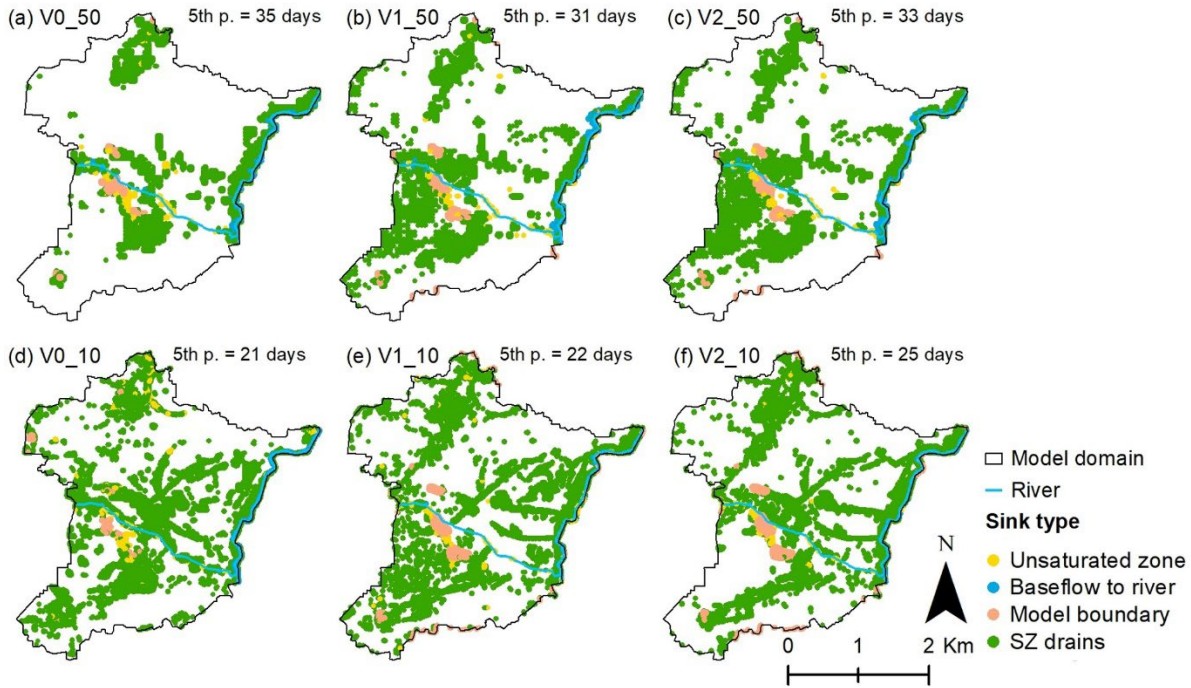

**Figure 13. Release locations of particles that are within the 5th percentile of the travel times to sinks and their end sink types. SZ drains stand for saturated zone drains and includes leaking sewers.**

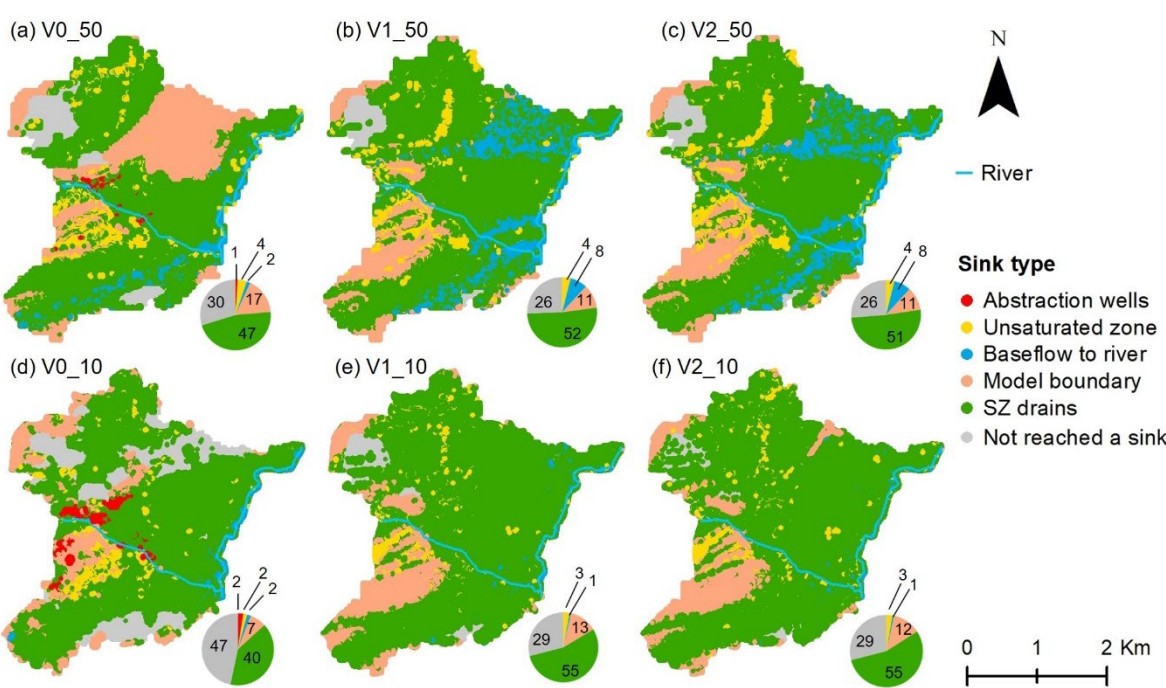

**Figure 14. Start location of all released particles and their end sink. SZ drains stand for saturated zone drains and includes leaking sewers. Pie diagrams show the percentage of particles ending up in the different sinks.**

In Figure 14 the starting locations for all particles released within the model area are shown and divided into the different sinks the particles discharge to (opposite Figure 13 where only the fastest 5[th] percentile of particles are shown). Pie diagrams of the percentages of the particles captured by the different sinks are shown as inserts in the figure.

The results presented in Figure 14 and Table 4 indicate that the urban geology incorporated in V1 and V2 introduces major changes in the groundwater pathways. Here, a higher percentage of the particles is captured by the drainage system, while the horizontal spatial discretization also impacts the distribution of particles among the sinks. In all models, 25-30% of the particles do not reach a sink within 200 years, except for V0_10 where 47% of the particles do not reach a sink. As seen in Figure 14, it is only in the V0 models that the released particles were captured by the abstraction wells.

## 5 Discussion

We tested the impact of urban geology for two discretizations; a coarse discretization and a finer discretization close to the scale of the urban subsurface infrastructure. The effect of the geological configuration and horizontal discretization was analyzed based on their simulation of high-water levels, the 95[th] percentile, and particle tracking. The level of detail in the voxel models depended on both the available data and their quality as well as the spatial discretization of the model. The knowledge of the properties of the infill, as well as the dimensions (e.g. exact size of trenches) of the anthropogenic structures, was limited, nevertheless, 7% of the volume in the voxel models was modified compared to the base model V0.

The calibration of the six hydrological models resulted in an almost similar fit to the observation data both in terms of mean error for head and stream discharge and in terms of the yearly head dynamics, see Figure 7 and 8. Yet, for the V1_10 and V2_10 the RMSE_h was 20 cm lower than for the other models. The better fit is most likely due to the better description and resolution of the shallow geology. In the objective function for the hydraulic head, all measurements were given the same weight, yet the RMSE_h of the shallow wells was generally better than for the deeper wells. The RMSE_h for the V1 and V2 models in 10 m resolution was 1.7 m and 1.0 m, respectively, for the deep and shallow wells.

By using preferred value regularization in the calibration process the parameter uncertainty was reduced. We considered the preferred parameter values to be realistic and therefore the parameter optimization was constrained such that unrealistic parameter estimates were avoided for the parameters that were less informed by observations as argued by Doherty (2015). The effect of the regularization was evident for, e.g., the root depth of grass ($D_{r,grass}$) for which unrealistic estimates were obtained without regularization. The parameter estimation with preferred value regularization thus constrained the estimation of parameters, which were poorly informed by the observation data, even though the relative error variance for these parameters was not reduced. Moreover, as a side effect of the regularization the resulting six models remained comparable in the sense that the parameter values remain within the same order of magnitude across the different models, see Figure 10.

The largest contrast in terms of estimated parameter values was found between the V0 model and the two urban models. The results from the parameter identifiability analysis showed that the parameters, which were more informed by observations changed from the base model V0 to the two urban models. This is especially clear for parameter $K_{Qs2,h}$, which showed fairly

warm colors in the V0 model while cold colors are dominating in the V1 and V2 models. This is mainly because of the sand/clay classes, which replaced much of the quaternary sand layers, e.g. KQs.

Furthermore, the analysis of the resulting parameter estimates, and their identifiability indicates that when the models are run with a finer horizontal discretization (10 m) the models become more sensitive to the hydraulic conductivity parameters, which have either high or low conductivity even though they are only locally present in the model. The lower sensitivity towards geological heterogeneity when the models represent a coarser spatial discretization is presumably because of their inability to resolve the heterogeneity.

The urban hydrological models with geology from V1 and V2 perform equally well, in both 10 and 50 m discretization, and similar groundwater levels and travel times were obtained. However, including information from geotechnical boreholes and soil data, did not lead to a noticeable change in the hydrological simulations. This could be because the shallow groundwater levels primarily were simulated to be located deeper than one meter below the terrain, which was the depth to which the soil map was represented. The largest change in hydrological simulations occurred from V0 to V1, when the urban infrastructure

data was included in the geological model, particularly utility trenches and road and railways build-up materials. The shallow basements (simulated as impermeable units) did not show a noticeable impact on the simulation. Only the very deep hospital basement (10 m) impacted the simulation of the depth to the water table, Figure 12. The reason why urban infrastructure has a significant impact on the hydraulic head is expected to be explained by the interconnection created between roads and utility trenches, coupled with the relatively high hydraulic conductivity of the build-up materials

compared to the surrounding geological settings. The particle tracking results from both V1_10, V1_50, V2_10, and V2_50 indicate that these conduits with high hydraulic conductivity create local preferential flow paths.

It was assumed that the entire sewer system was leaky and thus acted as a drainage system as well. A consequence of this was that the drains in the saturated zone were by far the most dominant sink to the shallow groundwater, see Figure 12 and Figure 13. Although this assumption about leaky sewers may not be correct in all areas of the model, groundwater seeping

into the sewers is a common problem and leads to treatment of excessive quantities of water in areas with shallow groundwater (Bhaskar et al., 2015; Rasmussen et al., 2022). If the pipe network in the model area is partly renovated, the water table may rise and reach unrenovated household drains that feed into the central storm or sewer network.

The horizontal discretization affected the hydrological simulation results, both for the groundwater levels and the particle tracking. The 95[th] percentile of the simulated water table depth by the six models showed that local variations in the

hydraulic head were smeared out in the 50 m models as compared to the variations in the 10 m models. Meanwhile, the urban models V1_10 and V2_10 provide a much more refined spatial simulation compared to both V0_10 and V0_50 models. This suggests that a hydrological model with a fine discretization can benefit from a detailed urban geological model, while a hydrological model with a discretization coarser than the scale of the subsurface infrastructures smears out the effect of local flow pathways. This is in agreement with Hibbs and Sharp (2012) who also found that a fine spatial

discretization is required to effectively utilize data on subsurface constructions such as trenches of water and sewer pipes, storm drains and other utility systems and hereby consider fast flow conduits generated by such structures.

The particle tracking showed that in models V1_10, V1_50, V2_10, and V2_50, recharge did not reach the deeper groundwater system. Rather, in these models with urban geology the shallow groundwater has near-surface flow paths and discharges to rivers, urban drainage systems, ponds, and lakes, see Figure 13 and Figure 14. This is as previously stated a
result of the interconnected subsurface infrastructures that cause preferential flow paths.

Comparing the particle tracking results across geological conceptualizations and spatial discretization it appears that the former is more important than the latter, and the results thus suggest that urban geology affects groundwater recharge. This has also been documented by Fletcher et al. (2013) among others. However, in these studies, the groundwater table is located below the sewer system and therefore receives indirect recharge from leaky water pipes, while in this study it was found that
the inclusion of the urban geology reduced recharge to deeper groundwater aquifers, as in the study by Bhaskar et al. (2015). This could have both a negative and positive effect on the deeper groundwater; negative in the sense that less recharge is available for the deeper groundwater resources, but positive in the sense that the deeper groundwater is not impacted by potentially contaminated water in shallow aquifers below the city.

The two sets of models with different spatial discretization produced the same results on simulated groundwater levels,
however, the 50 m resolution models simulated longer and very different residence times than the 10 models. This illustrates that the computational discretization affects the simulation of particle transport and travel time. The results suggest that a coarser spatial discretization is sufficient if it is only the groundwater level that is of interest, while for the simulation of groundwater ages and transport in a complex urban subsurface, a finer grid is required. These results are of importance for future studies of urban hydrogeology since the computational time of the two tested discretizations varied substantially. The
50 m models were 60 times faster in computational time than when the models had a 10 m horizontal discretization.

Mielby and Sandersen (2017) have made some suggestions on which resolution and discretization to use for different-sized urban models. They suggest that the spatial discretization of the model should be coarser when the size of the model increase and wise versa, which is a common approach to overcome the time and computational burden of hydrological models. Yet, if one wants to look into flow paths and to travel time of compounds moving between the surface water, groundwater, and
water infrastructures in urban areas, such as water extraction and leaking sewers and water pipes, the spatial resolution of the model input and the model discretization need to reflex the urban geology, such as pipes and trenches by keeping the discretization at a scale that does not smear out the heterogeneity of anthropogenic layer.

Moreover, as Salvadore et al. (2015) illustrate, the processes of sewer and mains leakage occur at a time scale of minutes to hours and at the resolution of cm to a few meters, whilst the surface water processes occur at 1 m to 1 km at a time scale of
seconds to hours. The groundwater processes occur at scales from less than one meter to kilometers and at many different timescales from an hour to years. Thus, this challenges the models' flexibility to integrate the dynamic processes at various spatial and temporal scales to produce realistic urban hydrological simulations of the groundwater level and flow paths in urban areas.

Solutions for the spatial discretization could be to have a regional model that can be updated and refined for a smaller area,
as done in this study and suggested by Mielby and Sandersen (2017). A drawback of constructing a model for a small aerial

extent is that it neglects the impact that local alterations can have on larger scales as documented in Attard et al. (2016a, c). Secondly, an alternative to the refinement method of this study could be to use a model where the spatial discretization can vary within the model domain (e.g., MODFLOW6), and thus have a finer discretization in the vicinity of the water infrastructure to capture the exchange that occurs in these places at short temporal intervals. Thirdly, one could use machine

learning as an alternative to or as an addition to a physical-based model. Koch et al. (2021) used simulation results from a coarse physically based model to guide a machine-learning model for the prediction of extreme groundwater levels at a finer scale. Meanwhile, Schneider et al. (2022) used a coarse physical-based regional hydrological model and ran smaller submodels at a finer scale to downscale the hydrological model outputs of climate change impact on the groundwater level for the regional model area to the finer scale. None of the studies were specifically focused on cities, yet the methods could

potentially be adapted to urban areas. The second and third suggestions are approaches that can be used for large-scale urban areas, and still have a fine spatial discretization near the urban anthropogenic elements.

Regarding the challenges of the different temporal resolutions of the processes, this study applied the MIKE She code, where it is possible to apply different computational timesteps for the three main components: surface, unsaturated zone, and saturated zone. This is also possible with other model codes such as PARFLOW.CLM as in Bhaskar et al. (2015) or

MODFLOW 6 (Langevin et al., 2017).

Ultimately, models are a reflection of the knowledge one has about the system and the data that is available for input, calibration and validation ((Madsen et al., 2010; Refsgaard et al., 2022). For urban studies, it is not only knowledge and data on climate, geography, geology, and hydrological data that is essential, but also data on the subsurface constructions, the backfill surrounding these constructions, utility trenches, water infrastructures, and human–water interactions. For instance,

one of the important model assumptions for this study is that leaky sewers will drain shallow groundwater. Here, the study is limited because measured groundwater inflow to the sewers is not available. Measurements of sewer flow and separation of the different inputs to the sewer system could qualify and verify this model assumption.

Since aging and leaking sewers and water pipes cause an impact on the water flow in urban areas it is relevant to document both new and existing subsurface infrastructure as the cities develop and to maintain these water infrastructures to ensure the

cities' resilience to both extreme weather conditions and slower, yet, progressively altering climate conditions (Hibbs and Sharp, 2012). This study had quite detailed information on the location, age, the extent of the subsurface constructions, and water infrastructure that will not always be available. Yet, as more and more constructions become digitally documented this data can be fed into the urban hydrological models, which can be updated with the new information. This study used GIS data, with properties of the pipes assigned in an attribute table and standard guidelines for dimensions of the trenches that the

pipes are placed in and the back material surrounding them. 3D building information modeling (BIM) is increasingly used for the design and documentation of buildings and can potentially more easily be combined with urban hydrological models. Nonetheless, reuse and updates of urban hydrological models are highly relevant since urban areas are frequently changing and since it requires many man-hours to build detailed integrated models.

## 6 Conclusion

This study examined the impact of anthropogenic urban geology and spatial discretization on the simulation of shallow groundwater at the city scale. The sensitivity of geological detail was analyzed by applying three geological models as input to three otherwise identical hydrological models, while the effect of spatial discretization was analyzed by varying the horizontal discretization of the hydrological models.

All the models were calibrated individually against the head observations from both shallow groundwater, deep aquifers, and

stream discharge with the use of preferred value regularization. This gave a good match to the observation data set for all models, while thecalibration with regularization constrained the parameters that were less informed by the observation data, to achieve reasonable parameter valueswhich were in the same order of magnitude across the six hydrological model versions.

The results of the stepwise development of the geological model and the test of its effect on the hydrological simulation

showed that the representation of the anthropogenic urban geology did not alter the depth of the water table to a large degree, yet it impacted the flow paths and travel time of the shallow groundwater. In terms of geological detail, the results showed that the largest impact on the simulation of the shallow groundwater was caused by the subsurface infrastructures; specifically, the utility trenches and the build-up material of roads and railways since these interconnected structures acted as preferential pathways for the shallow groundwater. Since the hydrological models assumed a leaky sewer system, the results

from this study suggest that the presence of these preferential flow paths also impacted the simulated recharge to deeper groundwater aquifers. The sewer system became a much larger sink of groundwater when these preferential flow paths were represented in the hydrological model.

In terms of spatial discretization, the largest impact of representing the urban geology was seen in the finer discretization of 10 m of the hydrological model compared to the 50 m discretization of the hydrological model, since the model in 50 m

discretization smeared out the effect of the local subsurface infrastructure preferential flow paths. Thus, the results from this study indicate that it is important to consider the dimension and hydrological properties of the urban subsurface infrastructures and subsurface buildings as they affect the local groundwater levels and the flow paths and hence also the sinks and recharge of the water to deeper aquifers and particles residence time in the aquifers.

In this study, a difference between the baseline model V0 and the models with detailed urban geology V1 and V2 was

evident despite the anthropogenic geology only representing 7 % of the total voxel model volume. This indicates that even though urban structures only alter a fraction of the shallow surface it may cause a significant impact on the hydrogeology which can change water levels, flow paths, end sinks, and travel time distributions to the sinks.

To simulate the water fluxes and flow paths in cities, the results from this study suggest that urban hydrological models need to include overland and subsurface drainage, as well as the water's interactions with water infrastructure, such as pumping,

drainage, and leaking sewers. Moreover, the results showed that the properties of the anthropogenic layer and the horizontal discretization of the shallow urban geology that represents the presence of infrastructure trenches, water infrastructure, and

subsurface constructions, need to be chosen carefully since these factors impact the model results as well as the computational time. To meet this computational and time-consuming challenge we propose that urban hydrological modeling consider the reuse of a large-scale regional model for setting up a refined urban hydrological model for a smaller

area, as in this study. Alternatively, for larger urban areas, we propose to vary the spatial discretization within the model domain, such that a smaller discretization is applied in the areas where infrastructure trenches, water infrastructure, and subsurface constructions are present, and potentially to make use of coupling machine-learning with an integrated physically based hydrological model.

*Author contribution:* AL and JK designed and executed the hydrological modeling work. AL led the data analysis and wrote the initial draft of the paper. MHM and PS designed and executed the geological modeling, KHJ, TOS and JK contributed scientifically to the modeling and data analysis. All authors contributed to the paper by providing comments, editing, and suggestions.

*Competing interests:* The authors declare that they have no conflict of interest.

*Acknowledgments:* The work was carried out as part of the SUBWATER project with financial support granted by the Danish Geocenter, VandCenter Syd A/S, Southern Region of Denmark, and the Municipality of Odense.

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
