# Peer review of "Impact of urban geology on model simulations of shallow groundwater levels and flow paths"

_Hydrology and Earth System Sciences, 2022_

## Author Comment (AC2)

**Reply to RC2: 'Comment on hess-2022-330'**

[Reviewer comments in normal font; *Author replies in italic*]

This paper examines the impact of representation of anthropogenic urban geology and spatial resolution on the simulation of shallow groundwater levels and flows. The authors developed two geological models from an existing hydrostratigraphical model by accounting for urban subsurface infrastructure and soil material and integrated them with hydrological models of 50m and 10m resolution. The effect of geologic configuration and spatial resolution are then analyzed in terms of high-water levels and particle tracking using a case study of the city of Odense in Denmark. Overall, I find the paper well written and comprehensive. It is worthy of publication.

*Reply: We thank the reviewer for the overall positive and constructive feedback to our work. Below, We will address the reviewer's comments and how we intend to respond to the issues pointed out by the reviewer.*

The methods and results are well-explained, and I have no comments on them. Since this paper is submitted as part of a special issue ('Representation of water infrastructures in large-scale hydrological and Earth system models'), I would suggest that the authors revise the introduction, discussion, and/or conclusions to bring out the broader implications of advancing representations of human-water interactions in hydrological/geological models due to increased impacts of anthropogenic interventions.

*Reply: The main motivation of the study was to test a method of representing urban geology at the city scale and to simulate the impact of anthropogenic interventions on the interactions between the hydrological system and the engineered water infrastructures. We believe that we have already discussed this topic in the introduction, discussion, and conclusions but in the revision process, we will strive to highlight and elaborate this even further as required.*

Other comments I have are minor issues to be corrected:
Line 260: Missing reference

*Reply: The missing reference in line 260 is Table 2. This will be corrected during the revision of the manuscript.*

Line 301-302: Consider revising to "and another set of parameters selected to be tied to…"

*Reply: This will be revised as required.*

*The current sentence in line 301-302:*

*Based on the analyses across all models a set of free parameters was selected subject to calibration and a set of parameters to be tied to the free parameters.*

*The sentence will be revised as required to:*

*Based on the analyses across all models a set of free parameters was selected subject to calibration and another set of parameters to be tied to the free parameters.*

Line 302: "and" instead of "sand"
*Reply: We will change the text as suggested by the reviewer.*

Line 315: "Eq" instead of "Ep"
*Reply: We will change the text as suggested by the reviewer.*

Line 317: "Eq. 2" instead of "Eq. 3.2"
*Reply: We will change the text as suggested by the reviewer.*

Eq. 2 and Line 325: Why do the weights have subscript hi, dj, and hk? Do they change according to indices i, j, k? Perhaps it will be clearer to specify the weights in Line 325 (e.g. state "$w_{hi}$ = 0.45")

*Reply: First, we would like to correct a mistake in the notation of the mathematical expression of the measurement objective function, Eq. 2.*

*The correct expression is:*

$$\Phi_m = \alpha_h * \sum_{i=1}^{h}(\omega_{h,i}(h_{obs,i} - h_{sim,i}))^2 + \alpha_{ampl} * \sum_{j=1}^{ampl}(\omega_{ampl,j}(ampl_{obs,j} - ampl_{sim,j}))^2 + \alpha_d * \sum_{k=1}^{d}(\omega_{d,k}(d_{obs,k} - d_{sim,k}))^2$$

$\alpha$ *is the group weight. The group weight was assigned as follows:*

$\alpha_h = 0.45, \ \alpha_{ampl} = 0.45, and \ \alpha_d = 0.10$

$\omega$ *is the weight of i'th, j'th or k'th observation. As a standard, all observations had a weight of 1, yet after initial calibration runs some of the head and amplitude observations located near the west boundary were assigned a value of 0.*

*During the revision of the manuscript, we will correct the mistake in Eq. (2) and specify the group weight more clearly in the text as suggested by the reviewer.*

Fig. 4: Refrain from using a rainbow colour scale as it could misrepresent data due to its non-linear change in hue
*Reply: We will change the color scale to be both a perceptually uniform scale and color-blind friendly.*

Fig 5: "56%" instead of "57%" in the bar plot

*Reply: Thank you for pointing out the inconsistency in the text and the numbers in Fig. 5. The correct value is 57% this will be corrected in the text of the manuscript during revision.*

Line 438 – Line 441: Please check the subscripts of the model parameters. Some capitalisations are inconsistent with those in Fig. 10

*Reply: The errors in the abbreviation for the parameter will be corrected as suggested by the reviewer during the revision of the manuscript:*

*In line 436: the parameter abbreviation will be corrected from Kqbv2,h and Kqc,v to $K_{bv2,h}$ and $K_{Qc,v}$*

*In line 437: the parameter abbreviation will be corrected from Kj25,h and dr,grass to $K_{cl,h}$ and $D_{r,grass}$*

*In line 438: the parameter abbreviation will be corrected from Kqbv2,h, Kqc,v , Kqs2,h and dr,grass to $K_{bv2,h}$, $K_{Qc,v}$, $K_{Qs2,h}$ and $D_{r,grass}$*

*In line 439: the parameter abbreviation will be corrected from Kqbv2,h to $K_{bv2,h}$.*

*In line 441: the model parameter will be corrected from Ks70,h, to $K_{S70,h}$.*

Line 446-447: delete "4.3 Simulation of high-water levels."
*Reply: Thank you for pointing out this typo error. We will delete this during the revision of the manuscript.*

Line 452: "V2_50" instead of "V2_5"

*Reply: We will change the text as suggested by the reviewer.*

Line 477: add "models" after "V2"

*Reply: We will change the text as suggested by the reviewer.*

Line 509, 517: Check capitalization and subscript of parameter "$D_{r,grass}$" and "$K_{Qs2,h}$"

*Reply: This will be corrected during the revision of the manuscript as suggested by the reviewer. In line 509: the parameter abbreviation will be corrected from dr,grass to $D_{r,grass}$.*

*In line 517: the parameter abbreviation will be corrected from $K_{qs2,h}$ to $K_{Qs2,h}$*

*In line 519: the parameter abbreviation will be corrected from $K_{qs}$ to $K_{Qs}$*

---

## Author Comment (AC3)

**Reply to RC3: 'Comment on hess-2022-330'**

[Reviewer comments in normal font; *Author replies in italic*]

This manuscript deals with the modelling of shallow groundwater flows and levels in urbanized catchments, and highlights the impact of both the urban geology description and the spatial resolution used in the distributed model. This topic is of high interest, because the interactions between groundwater and underground constructions are important in urban soils whose features are very variable, and we need to improve our ability to simulate these complex hydrological behaviours.

The study is based on an integrated hydrological model using MIKE SHE code and this model allows a detailed representation of groundwater levels and flows. Velocity fields and then travel times may be deduced from the model; this is a real added value of this modelling application : this type of result is quite rare in the field of urban groundwater modelling and it has to be noticed. The impact of urban infrastructures in the shallow groundwater flows and level is proved through this study and this is a step forward in the urban hydrology behaviour knowledge.

The structure of the paper is basic and clear, with a first introduction section presenting the main issues related with this topic and a short state of the art dealing with urban shallow groundwater modelling, and a focus on the importance of the soil and geology description. The second section includes the case study presentation. The Geological models and the main modelling methodology adopted here is presented then and the data- modelling- and evaluation methodology adopted here. The last sections are usual, with results, discussion and conclusion.

*Reply: We thank the reviewer for the overall positive and constructive feedback on our work. Below, We will address the reviewer's comments and how we intend to respond to the issues pointed out by the reviewer.*

**General opinion and minor comments**

This manuscript is devoted to the sensitivity of an integrated hydrological model to the urban geology, and uses 3 different representations (i.e. 3 geological models) with various consideration of the specific urban soil features. The sensitivity of the model to the spatial resolution is analysed too. For this last factor, I wonder if only two grid sizes is enough for the study of the effect of the spatial resolution.

*Reply: We acknowledge that by only testing two grid sizes for the hydrological model we cannot claim to have done an exhausting analysis of the sensitivity of the model to the spatial resolution. Furthermore, as we have written to the first reviewer, we concede that the manuscript lacks a justification for the choice of discretization of both the geological voxel models and the hydrological models.*

*As written in the response to the first reviewer we suggest to add the following lines in the method section:*

*Text to add after line 160:*

*The effect of spatial discretization was tested by using a coarse discretization relative to the urban subsurface infrastructure and a finer discretization close to the scale of the urban subsurface infrastructure, e.g., roads and trenches. For both the geological and hydrological modeling tools a discretization in the order of 1-10 m becomes computationally challenging when the size of the model is large, that is millions of grid cells.*

*Text to add after line 180:*

*The choice of discretization for the voxel model of urban geology was guided by the experience from the study of Mielby & Henriksen (2020;) and Mielby & Sandersen (2017) and chosen to be a 5 x 5 x 1 m to be able to represent the subsurface cable trenches which are typically 1-3 m wide and 1-2 m in depth for this study area. The horizontal discretization was thus larger than the dimensions of the trenches. This was because a smaller discretization for a model at the city scale would have been too computationally heavy for the hydrological model to handle. Mielby & Sandersen (2017) argued that the discretization of the geological and hydrological model has to meet the required detail, yet not exceed the computational capabilities. The two voxel models each had 22 million voxel grids.*

*Text to add after line 212:*

*For the hydrological modeling, multiple grid sizes were initially tested. The two grid sizes of 50 and 10 m were chosen based on a tradeoff between computation time and the number of grid cells for the model size.*

*Furthermore, we will elaborate on the topic of choice of grid sizes for urban hydrogeological models in the discussion section during the revision of the manuscript.*

*A suggestion for text to add to the discussion during the revision of the manuscript:*

*We wanted to test the impact of urban geology for only two discretizations; a coarse discretization relative to the urban subsurface infrastructure and a finer discretization close to the scale of the urban subsurface infrastructure, e.g., roads and trenches.*

*The two sets of models with different spatial discretization produced the same results on simulated groundwater levels, however, the 50 m resolution models simulated longer residence time than the 10 models. These results illustrate that the computational grid size can affect the simulation of particle transport and residence time.*

*These results are of importance for future studies of urban hydrogeology since the computational time of the two tested resolutions varies substantially. The 50 m models were 60 times faster in computational time than the 10 m resolution models. A coarser grid resolution is thus only sufficient if the groundwater level dynamics are of interest, while for the simulation of groundwater ages and transport in a complex urban subsurface, a finer grid is required.*

Minor comments

The overall manuscript, including methods and results, is relevant and well-prepared and written. However, I have a few minor comments that could be into account in order to improve the quality of the manuscript and help the reader.

First of all, I noticed a lack of justification, especially in the Methods section. The authors did not always argue their assumptions :

- p5 l 118 : "… concrete pavement , which have an imperviousness of 75%" . How was this value estimated? Traditionally, this kind of surface is considered as totally impervious. But I acknowledge that it may be partially pervious. But that should be explained.

*Reply: The quoted sentence says above 75% in the manuscript. The sentence is a description of the map in Figure 1b, which shows the imperviousness in percentage in 10 m grids. The data on imperviousness is from a raster map from the Danish Geodata Agency (2019).*

*Buildings and pavements are as the reviewer points out normally considered 100% impervious. Yet, the map contains areas where the imperviousness is 75%. This can be places where a little area with vegetation is placed next to a building or a road.*

*As suggested by the reviewer, we will specify this in the manuscript during revision.*

*In the manuscript, in lines 116-118 the text says:*

*"Approximately 50 % of the total land cover in the model domain is impermeable or semi-permeable such as buildings, asphalt, and concrete pavement, which have an imperviousness above 75 %."*

*We propose to add the following lines after this sentence:*

*Buildings and pavements are normally considered 100% impervious. Yet, the map contains areas where the imperviousness is 75%. This can be places where a little area with vegetation is placed next to a building or a road.*

-p7 l 183 : " … and additional data on soil material in the top 5 meters". As the modelling application is quite sensitive to the soil configuration, especially in the first meters, one can wonder where this "additional data" comes from! What kind of additional data? From drilling data? From infiltration tests?

*Reply: As suggested by the reviewer,  we will specify what data was used on soil material during the manuscript revision.*

*In the sentence that follows the cited line 183, we refer to table S1 which presents an overview of the data for the geological models: "The different data and sources utilized for the geological models are presented in the supplementary material (Table S1). ".*

**Table S1. Overview of data for the geological models**

| Category | Data type | Source | The geological model that utilized the data |
|---|---|---|---|
| Hydrostratigraphical model (V0) | 3D layer model in GeoScene | Sandersen and Kallesøe (2017) | V0, V1, V2 |
| Urban infrastructure | Sewer network | Vandcenter Syd A/S (2019a) | V1, V2 |
| | Water supply pipes | Vandcenter Syd A/S (2019b) | V1, V2 |
| | District heating pipes | Fjernvarme Fyn (2019) | V1, V2 |
| | Gas pipes | Danish Gas Distribution (2019) | V1, V2 |
| | Roads and railways | Danish Geodata Agency (2019) | V1, V2 |
| | Road build-up material | Vejdirektiratet (2019) | V1, V2 |
| | Railway build-up material | Nielsen (2016) | V1, V2 |
| | Buildings and basements | Odense Kommune (2019) | V1, V2 |
| Soil material | Soil mail | Jakobsen et al. (2022) | V2 |
| | Shallow geotechnical boreholes | Geological surveys of Denmark and Greenland (GEUS) (2019) | V2 |

*We propose to add the following lines to the manuscript after the reference to the table in the supplementary material:*

*The additional data on soil material in V2 voxel model is a soil map (Jacobsen et al. 2022) and soil descriptions from shallow geotechnical boreholes (GEUS, 2019). The soil map by Jacobsen et al. (2022) is in 1:25000 resolution and is based on samples of soils every 200 m at 1 m depth. The soil descriptions from shallow geotechnical boreholes were derived by looking through non-digitalized documents in the Danish National well database.*

-P8 l 207-209 " the location of roads and pipes (…) were used as proxies for the presence of excavations and trenches" What is the relevance of this assumption? Did you assess this assumption? Did you compare this proxies methodology to real data? Is it valuable only in this study case or could it be transposed in any urban catchment?

*Reply: We acknowledge that the quoted sentence is a vague formulation of the methodology of defining the extent of infrastructure in the geological voxel models. We propose to add the following lines to the manuscript in line 208 after the quoted sentence:*

*The location of the roads and the pipes were retrieved from the road directory and the pipe owners, see table S1 for data sources. The extent of the excavations and trenches was based on national standards for profiles of road design and pipe trenches, see table S1 for sources. It was assumed that the design of the roads, railways, and trenches followed these standards.*

*To answer the reviewer's questions we find that this method of proxies for the presence of excavations and trenches is the best possible way unless the extent of the trenches is documented and stored in a central and digitalized archive. This is not the practice for this study area and we suspect it is rarely the case for other cities. Moreover, to answer the last question from the reviewer, the presented methodology can be transposed to other urban catchments.*

- P8 214-220 – Why the SHE model was chosen here? We can understand that it is the model used by the research team, but could the authors argue why this model is appropriate to do this study? Are there any equivalent modelling tools/methods that could have been considered for this type of modelling study? Is SHE model the only one that allows to achieve the objectives of this study?

*Reply: We acknowledge that the manuscript lacks an argumentation for the choice of model code. We propose to add the following to the manuscript in the beginning of the method section 3.2 Hydrological models (line 212):*

*THE MIKE SHE model was chosen for the hydrological simulation because the model can simulate the surface and subsurface processes in an integrated and dynamic manner, as well as it is possible to include surface and sewer drainage. An advantage of MIKE SHE is that it can describe the water flow between different types of surfaces, the root zone, the unsaturated zone, and the saturated zone. Moreover, with this model code, the properties, surfaces, and layers can be spatially distributed in both the horizontal and vertical planes. Other integrated models such as PARFLOW.CLM, MODFLOW 6, and HydroGeoSphere offer similar capabilities, however, given our experience with MIKE SHE we selected this model.*

- p9 l 245. What is this surface-subsurface leakage coefficient? A parameter of the SHE model? Does it take into account the leakage in pipes, or only the leakage from surface-subsurface? How could it be estimated?

*Reply: The surface-subsurface leakage coefficient is a model parameter in MIKE SHE. It reduces the infiltration from the surface to the subsurface at paved surfaces as well as the seepage from the subsurface to the surface. In the model, it is applied to the areas where the imperviousness is above 50%.*

*Surface-subsurface leakage coefficient does not account for leakage in pipes. Leakage in the pipes was modeled separately and it was assumed that leakage only occurs to the sewer pipes. The leakage was modeled by representing the sewer network as subsurface drainpipes and assuming that the parameter was spatially uniform across the pipe network.*

*We will make this clear in the revised manuscript.*

Then, the methods section could have been improved with a graphical scheme helping the reader to understand the chosen parametrizations. This is especially needed in the 3.2.2 paragraph, because the list of the presentation of the parametrization and boundary conditions is quite long, and a scheme would be more efficient and more easy for the reader.

*Reply: The parameterization is indeed complicated. Some parameters are defined from data, some were specified from past model experience and some were subject to calibration as described in section 3.3.*

*To enhance the readability of the parameterization we suggest to expand the revised manuscript with either an additional table or a figure.*

*A figure could be an illustration of the model setup such as below with an additional description of the reduced infiltration (the surface-subsurface leakage coefficient ), the overland drainage component, and leaking sewers:*

[Figure]

Finally, I have a short comment about one element of discussion : l 535-543. The sewers renovation could be a way to reduce the soil-sewer interactions and the infiltration of groundwater in sewers. As discussed by the authors, the preferential flow paths would still be present in the pipe trenches. However, I wonder if having a full renovated sewer system is not an utopy… To my opinion, there will still be some defects in the sewer system and then, as the preferential flow in the trenches remains present, the water will always find a way to penetrate in the sewers. I have the impression that this type of sewer renovation (or "non leaking pipes assumption") is only a "modelling dream"; I am not sure it would be feasible in reality.. (especially in a economical point of view). I would appreciate that the authors re-consider this paragraph.

*Reply: We agree that it is probably not realistic to install a leakage-free pipe network. As the reviewer correctly states, the preferential flow in the trenches remains present, and the water will always find a way to the leaking sewers. In the discussion of the revised manuscript, we speculate*

*on the possible impact of the renovation of sewers and we will expand on this topic in the revised manuscript.*

*We propose to rephrase lines 535-543 in the discussion to:*

*It was assumed that the entire sewer system was leaky and thus acted as a drainage system as well. In consequence, drains in the saturated zone were by far the most dominant sink to the shallow groundwater, Figure 12 and Figure 13. Although this assumption may be on the extreme side, groundwater seeping into the sewers is a common problem and leads to excessive water treatment in areas with shallow groundwater. On the other hand, in an idealized case where sewers have been renovated, the water table may raise and trigger water seeping into basements or a periodical groundwater table above the terrain. Yet, a fully renovated pipe network across an urban area is unrealistic. The groundwater will even if parts of the network were renovated still flow along the preferential flow paths and the water will just find another leak in the pipe network. Say e.g a part of the central pipe network was renovated, the water table may rise and reach the unrenovated household drains that feed into the central storm or sewer network.*

**References**
Several mistakes should be corrected :

*Reply: Thank you for pointing out these mistakes. We will correct these in the revised manuscript.*

- l57 Boukhemacha et al (2051) and Epting et al. (2008) are missing in the list of references

*Reply:*

*Boukhemacha, M. A., Gogu, C. R., Serpescu, I., Gaitanaru, D., and Bica, I.: A hydrogeological conceptual approach to study urban groundwater flow in Bucharest city, Romania, Hydrogeol. J., 23, 437–450, https://doi.org/10.1007/s10040-014-1220-3, 2015.*

*Epting, J., Huggenberger, P., and Rauber, M.: Integrated methods and scenario development for urban groundwater management and protection during tunnel road construction: A case study of urban hydrogeology in the city of Basel, Switzerland, Hydrogeol. J., 16, 575–591, https://doi.org/10.1007/s10040-007-0242-5, 2008.*

- l115 / l 633 : Danish Geodata Agency ?

*Reply: Danish Geodata Agency: FOT data, www.gst.dk, 2019.*

- l197 Kristensen et al (2015) is missing in the list of references

*Reply:*

*Sandersen, P. B. E., Kristensen, M., and Mielby, S.: Udvikling af en 3D geologisk/hydrogeologisk model som basis for det urbane vandkredsløb. Delrapport 4 - 3D geologisk/hydrostratigrafisk modellering (Særudgivelse)., De Nationale Geologiske Undersøgelser for Danmark og Grønland, Denmark, 2015.*

- l 227 DHI 2017 is missing

*Reply:*

*DHI: MIKE HYDRO River User guide, 2017.*

- l 260 is specified in Fejl ! ... Like fundet / to be corrected

*Reply:*

*Table 2: Conditions for the computational layers in the hydrological models*

---

## Author Response (AR1)

**Author's response to the revision of hess-2022-330**

**Editor Comment**

[Editor's comments in normal font; *Author's response in italics*]

Dear authors, thank you for the detailed response to the reviewers' comments. I suggest to proceed with a moderate revision aimed at implementing all points that have been raised, mostly related to modelling assumptions and experimental setup.

I also encourage the authors the fully address the comment raised Referee #2 on the representation of human-water interactions in hydrological/geological models (comment #2). It is true that the Introduction clearly outlines the goals of this study, but it is also true that other parts of the manuscript could be expanded to create a stronger tie with the overarching theme of this special issue ('Representation of water infrastructures in large-scale hydrological and Earth system models'). The Discussion, for instance, does an excellent job in discussing the specific results of this study, but it could be slightly expanded to elaborate on some of the open challenges (e.g., data availability, computational requirements, scalability to even larger domains, model availability) that will be faced when advancing the representation of urban geology in hydrological/geological models. I believe such expanded discussion will be interesting to the readership of this SI.

*Author's response:*
*Thank you for the opportunity to submit a revised manuscript based on the three reviewers' comments. We are very appreciative of the constructive suggestions to improve the manuscript. In the revised manuscript, we have implemented all points that have been raised and corrected typos.*
*Please note that all page and line numbers mentioned under Author's changes refer to the marked-up manuscript version.*

**RC1: 'Comment on hess-2022-330'**

[Reviewer's comments (RC) in normal font; *Author's response (AR)in italics*; Author's changes (AC) in manuscript]

RC1.1
Overall, this is an interesting manuscript worth being published.
I do see, however, some room for improvement, especially in the use of the terms "resolution" and "spatial discretization". The authors appear to use different spatial resolution (10m and 50m in the horizontal direction), both of which obviously come with a different degree of geological resolution. It is unclear whether the different model results are then attributed to the spatial discretization or to the different geological resolution. To disentangle this, a grid discretization study should be conducted first, and the same grid should then be used to examine the effect of different geological resolution. I think as is, the term spatial discretization is confused with geological resolution. A poor spatial discretization may result in round-off and truncation errors, which are numerical artefacts. A fine enough grid should be free of numerical artefacts, and this grid could indeed be used to test different degrees of geological resolution. Different geological resolution may actually show different results (just like with different spatial discretizations) but the effect of numerical artefacts would be absent.

*AR1.1:*

*We thank the reviewer for the overall positive and constructive feedback that has helped us to improve our work. Please note that all page and line numbers mentioned under Author's changes refer to the marked-up manuscript version.*

*We acknowledge that the terms "spatial resolution and "spatial discretization" are used inconsistently in the manuscript. One of the objectives was to analyze the effect of spatial discretization on the simulations. As part of the manuscript revision, we have replaced the word "resolution" with "discretization" since discretization is the term that best describes what we did when we chose a grid size of 5 x 5x 1 m for the voxel models and tested different horizontal grid sizes for the hydrological model.*

*Furthermore, the reviewer's comment also shows that it was unclear which discretization was used in the different geological and hydrological models. Besides a consistent use of terminology, we have included a table of the modeling experiments in the revised version of the manuscript.*

*The reviewer suggests adding an initial grid discretization analysis to the study. We agree that both the geological resolution and discretization can affect the simulation results. Rather than documenting a full grid analysis in the manuscript, we have added some lines to the method section that justifies our choice of grid discretization.*

AC1.1:

The word use of "resolution", "discretization" and "computational grid size" has been streamlined to assure proper use of terminology and consistency, e.g. in the revised manuscript page 1, line 9 "resolution" was replaced with "discretization".

In the material and methods section of the revised manuscript page 7, lines 168-179 the following text has been altered:

The effect of the geological configuration was analyzed by applying three geological models V0, V1, and V2 in otherwise identical  hydrological models.  The effect of spatial discretization was tested by using a coarse discretization relative to the urban subsurface infrastructure, with a horizontal grid discretization of 50 m, and a finer discretization close to the scale of the urban subsurface infrastructure, e.g., roads and trenches, with a horizontal grid discretization of 10 m. For both the geological and hydrological modeling tools a discretization in the order of 1-10 m becomes computationally challenging when the size of the model is large, resulting in e.g., millions of grid cells. For the hydrological modeling, multiple grid sizes were initially tested. The two grid sizes of 50 and 10 m were chosen based on a tradeoff between computation time and the number of grid cells for the model size and retaining geological detail. The effect of the geological configuration and spatial discretization  was analyzed based on the hydrological models' simulation of high-water levels, the 95th percentile, and particle tracking.

Furthermore, on page 7, lines 182-186 a new table has been added to the revised manuscript.:

Sandersen and Kallesøe (2017). The geological models V1 and V2 were developed as part of this study. An overview of the geological models and their differences is presented in Table 1.

**Table 1. Overview of models and their differences in discretization and geological model type**

| Hydrological model name | Horizontal discretization of the hydrological model (m) | Geological model name | Geological model type | Discretization of the geological model |
|---|---|---|---|---|
| V0_50 | 50 | V0 |  Layer model | Layers and lenses with varying thickness |
| V0_10 | 10 | | | |
| V1_50 | 50 | V1 |  Combined voxel and layer model with urban infrastructure | 5x5x1 m in the voxel model (the top 15 mbgl). Layers and lenses with varying thicknesses |
| V1_10 | 10 | | | |
| V2_50 | 50 | V2 |  Combined voxel and layer model with urban infrastructure and soil material | 5x5x1 m in the voxel model (the top 15 mbgl). Layers and lenses with varying thicknesses. |
| V2_10 | 10 | | | |

In the revised manuscript page 8, line 222-209 the following text has been added:

anthropogenic structures in the uppermost subsurface and computational constraints. The choice of discretization for the voxel models with urban geology was guided by the experience from the study of Mielby & Henriksen (2020) and Mielby & Sandersen (2017) and chosen to be 5x5x1 m to be able to represent the subsurface infrastructure trenches which are typically 1-3 m wide and 1-2 m in depth for this study area, while the Road trenches are around 10 m wide +/- and therefore 5 m is a good intermediate size. The horizontal discretization was thus larger than the dimensions of the trenches. A smaller discretization for a model at the city scale would have been computationally expensive in the hydrological model. Mielby & Sandersen (2017) argued that the discretization of the geological and hydrological model must meet the required detail, yet not exceed the computational capabilities. The two voxel models each had 22 million voxel grids.

RC1.2:

Maybe the authors could also re-think the title. It is unclear whether groundwater quantity is meant, or quality, or level, or availability? This should perhaps be clarified.

*AR1.2:*

*Concerning the title, we agree that the title should be more specific and suggest the following title: 'Impact of urban geology on model simulations of shallow groundwater levels and flow paths'*

AC1.2:

Page 1, lines 1-2 the title has been changed to:

Impact of urban geology on model simulations of shallow groundwater levels and flow paths

**RC2: 'Comment on hess-2022-330'**

[Reviewer's comments (RC) in normal font; *Author's response (AR)in italics*; Author's changes (AC) in manuscript]

RC2.1:

This paper examines the impact of representation of anthropogenic urban geology and spatial resolution on the simulation of shallow groundwater levels and flows. The authors developed two geological models from an existing hydrostratigraphical model by accounting for urban subsurface infrastructure and soil material and integrated them with hydrological models of 50m and 10m resolution. The effect of geologic configuration and spatial resolution are then analyzed in terms of high-water levels and particle tracking using a case study of the city of Odense in Denmark. Overall, I find the paper well written and comprehensive. It is worthy of publication.

*AR2.1:*

*We thank the reviewer for the overall positive and constructive feedback on our work. Below, We will address the reviewer's comments and what we have changed due to the issues pointed out by the reviewer. Please note that all page and line numbers mentioned under Author's changes refer to the marked-up manuscript version.*

RC2.2:

The methods and results are well-explained, and I have no comments on them. Since this paper is submitted as part of a special issue ('Representation of water infrastructures in large-scale hydrological and Earth system models'), I would suggest that the authors revise the introduction, discussion, and/or conclusions to bring out the broader implications of advancing representations of human-water interactions in hydrological/geological models due to increased impacts of anthropogenic interventions.

*AR2.2:*

*The main motivation of the study was to test a method of representing urban geology at the city scale and to simulate the impact of anthropogenic interventions on the interactions between the hydrological system and the engineered water infrastructures. We have slightly adjusted the introduction and expanded the discussion and conclusions in the revised manuscript to highlight and elaborate this even further as required.*

AC2.2:

In the revised manuscript page 2, lines 29-37, we have adjusted the opening paragraph to address the theme of the special issue.

As more than half of the world's population lives in urban areas and urbanization globally continues to increase (United Nations, 2018), urban water resources receive increasing attention (McGrane, 2016; Lundy and Wade, 2011; Mitchell, 2006; Farr et al., 2017; Birks et al., 2013).  Cities are hydrologically complex, because of interactions between  built structures,  water infrastructures, such as pumps, drainage, sewers and water pipes, and the natural hydrological system, where  surface and subsurface processes  occur at various spatial and temporal scales (Salvadore et al., 2015; Han et al., 2017; Kidmose et al., 2015; Fletcher et al., 2013; Tubau et al., 2017; Vázquez-Suñé et al., 2016).

In the revised manuscript page 29, lines 647-701, we have added the following to the discussion section in order to elaborate on the implications of advancing the representation of human-water interactions in hydrological models.

[revised manuscript text omitted]

In the revised manuscript page 32, lines 733-743, we have added the following to the conclusion section in order to elaborate on the implications of advancing the representation of human-water interactions in hydrological models.

To simulate the water fluxes and flow paths in cities, the results from this study suggest that urban hydrological models need to include  overland and subsurface drainage, as well as the water's interactions with water infrastructure, such as pumping, drainage, and leaking sewers. Moreover, the results showed that the properties of the anthropogenic layer and the horizontal discretization of the shallow urban geology that represents the presence of infrastructure trenches, water infrastructure, and subsurface constructions, need to be chosen carefully since these factors impact the model results as well as the computational time. To meet this computational and time-consuming challenge we propose that urban hydrological modeling consider the reuse of a large-scale regional model for setting up a refined urban hydrological model for a smaller area, as in this study. Alternatively, for larger urban areas, we propose to vary the spatial discretization within the model domain, such that a smaller discretization is applied in the areas where infrastructure trenches, water infrastructure, and subsurface constructions are present, and potentially to make use of coupling machine-learning with an integrated physically based hydrological model.

**Minor comments**

RC2.3

Line 260: Missing reference

*AR2.3: The missing reference in line 260 (original manuscript) is Table 2. This has been corrected during the revision of the manuscript, yet as one more table have been added to the manuscript the correct reference is now Table 3..*

AC2.3: In the revised manuscript the reference has been corrected on page 13, line 323.:

layers and their outer boundary conditions is specified in **Error! Reference source not found.**, and even though the geological models were altered the

RC2.4:

Line 301-302: Consider revising to "and another set of parameters selected to be tied to…"

*AR2.4:This has been revised as required.*

AC2.4: In the revised manuscript page 15, lines 364-366 the text has been changed to:

The sensitivity analysis was based on composite sensitivities as described in Doherty (2015)  and conducted for all 96 model parameters. Based on the analyses across all models a set of free parameters was selected subject to calibration and another set of parameters to be tied to the free parameters.

RC2.5:
Line 302: "and" instead of "sand"
*AR2.5: We have changed the text "sand" to "and" as suggested by the reviewer.*

AC2.5:  In the revised manuscript (page 15, line 366) the text has been changed to:

One parameter set was selected for the V0 model and one

RC2.6:
Line 315: "Eq" instead of "Ep"
*AR2.6: We have changed the text as suggested by the reviewer.*

AC2.6: In the revised manuscript (page 16, line 379) "Ep" has been changed to "Eq"

RC2.7:

Line 317: "Eq. 2" instead of "Eq. 3.2"

*AR2.7: We have changed the text as suggested by the reviewer.*

AC2.7: In the revised manuscript (page 16, line 381) "Eq. 3.2" has been replaced by "Eq.( 2)"

RC2.8:

Eq. 2 and Line 325: Why do the weights have subscript hi, dj, and hk? Do they change according to indices i, j, k? Perhaps it will be clearer to specify the weights in Line 325 (e.g. state "$w_{hi} = 0.45$")

*AR2.8:*

*First, we would like to correct a mistake in the notation of the mathematical expression of the measurement objective function, Eq. 2. During the revision of the manuscript, we have corrected the mistake in Eq. (2) and specified the group weight more clearly in the text as suggested by the reviewer.*

AR2.8: In the revised manuscript the (page 16, lines 385-392) have been changed to:

$$\Phi_m = \alpha_h * \sum_{i=1}^{h}(\omega_{h,i}(h_{obs,i} - h_{sim,i}))^2 + \alpha_{ampl} * \sum_{j=1}^{ampl}(\omega_{ampl,j}(ampl_{obs,j} - ampl_{sim,j}))^2 + \alpha_d *$$

$$\sum_{k=1}^{d}(\omega_{d,k}(d_{obs,k} - d_{sim,k}))^2 \qquad\qquad\qquad (2)$$

where α is the group weight. The group weight was assigned as follows: α_h=0.45, α_ampl=0.45, and α_d=0.10. ω_is the weight of i'th, j'th or k'th observation. As a standard, all observations had a weight of 1, yet after initial calibration runs some of the head and amplitude observations located near the west boundary were assigned a value of 0.

RC2.9:

Fig. 4: Refrain from using a rainbow colour scale as it could misrepresent data due to its non-linear change in hue.

*AR2.9: We have changed the color scale to be both a perceptually uniform scale and color-blind friendly.*

AC2.9:

The color of the figure (page 18, lines 438-440) has been changed as shown below. Due to a new figure in the manuscript the figure number has been changed from figure 4 to figure 5:

[Figure]

**Figure 1. Profile A-A' from the geological models V0 – layer model (a), V0 – voxel model, (b) V1 – voxel model (c), and V2 – voxel model (d).**

RC2.10:

Fig 5: "56%" instead of "57%" in the bar plot

*AR2.10: Thank you for pointing out the inconsistency in the text and the numbers in Fig. 5. The value has been corrected in the figure.*

AC2.9:

The figure (page 19, line 441-444) has been changed as shown below. Due to a new figure in manuscript the figure number have been changed from figure 5 to figure 6:

[Figure]

Figure 2. **Percentage of the sand/clay fractions of the total volume of the geological voxel models V0, V1, and V2. Note that the sand/clay fractions 0.6-0.7 and 0.9-1.0, respectively take up about 37 % and 56 % in all models, and plot beyond the range of the y-axis.**

RC2.11:

Line 438 – Line 441: Please check the subscripts of the model parameters. Some capitalisations are inconsistent with those in Fig. 10

*AR2.11: The errors in the abbreviation for the parameters has been corrected as suggested by the reviewer during the revision of the manuscript.*

AC2.11: The parameter abbreviations in the manuscript has been correct throughout the results, see e.g page 23, line 506-507:

conductivities $K_{qbv2,h}$ and $K_{Qc,v}$ are well informed by measurements since they are dominated by red-yellow colors. For V0_50 the parameters dominated by blue-green colors are $K_{ct,h}$ and $D_{r,grass}$, while for V0_10 it is $K_{ct,h}$ and $C_{riv}$.

RC2.12:
Line 446-447: delete "4.3 Simulation of high-water levels."

*AR2.12: Thank you for pointing out this typo error. We will delete this during the revision of the manuscript.*

AC2.12: In the revised manuscript the typo have been deleted at the end of the paragraph on page 23, line 517-518:

the regularized calibration were considered realistic and well-defined and were accepted for further analysis. 4.3 Simulation of high-water levels

RC2.13:
Line 452: "V2_50" instead of "V2_5"
*AR2.13: We have changed the text as suggested by the reviewer.*

AC2.13:In the revised manuscript (page 24, line 525) "V2_5" has been replaced with "V2_50"

RC2.14:
Line 477: add "models" after "V2"
*AR2.14:We have changed the text as suggested by the reviewer.*

AC2.14: In the revised manuscript (page 25, line 523) "models" has been added after "V2"

RC2.15:

Line 509, 517: Check capitalization and subscript of parameter "Dr,grass" and "KQs2,h"

*AR2.15 This will be corrected during the revision of the manuscript as suggested by the reviewer.*

AC2.15: In the revised manuscript page 27, line 583 the parameter abbreviation has been corrected from $d_{r,grass}$ to $D_{r,grass}$, and on page 28, line 591 from $K_{qs2,h}$ to $K_{Qs2,h,}$.

**RC3: 'Comment on hess-2022-330'**

[Reviewer's comments (RC) in normal font; *Author's response (AR)in italics*; Author's changes (AC) in manuscript]

RC3.1:

This manuscript deals with the modelling of shallow groundwater flows and levels in urbanized catchments, and highlights the impact of both the urban geology description and the spatial resolution used in the distributed model. This topic is of high interest, because the interactions between groundwater and underground constructions are important in urban soils whose features are very variable, and we need to improve our ability to simulate these complex hydrological behaviours.

The study is based on an integrated hydrological model using MIKE SHE code and this model allows a detailed representation of groundwater levels and flows. Velocity fields and then travel times may be deduced from the model; this is a real added value of this modelling application : this type of result is quite rare in the field of urban groundwater modelling and it has to be noticed. The impact of urban infrastructures in the shallow groundwater flows and level is proved through this study and this is a step forward in the urban hydrology behaviour knowledge.

The structure of the paper is basic and clear, with a first introduction section presenting the main issues related with this topic and a short state of the art dealing with urban shallow groundwater modelling, and a focus on the importance of the soil

and geology description. The second section includes the case study presentation. The Geological models and the main modelling methodology adopted here is presented then and the data- modelling- and evaluation methodology adopted here. The last sections are usual, with results, discussion and conclusion.

*AR3.1:*

*We thank the reviewer for the overall positive and constructive feedback on our work. Below, we respond to the reviewer's comments and specify what changes there have been made in the revised manuscript based on the reviewer's comments. Please note that all page and line numbers mentioned under Author's changes refer to the marked-up manuscript version.*

**General opinion and minor comments**

RC3.2:

This manuscript is devoted to the sensitivity of an integrated hydrological model to the urban geology, and uses 3 different representations (i.e. 3 geological models) with various consideration of the specific urban soil features. The sensitivity of the model to the spatial resolution is analysed too. For this last factor, I wonder if only two grid sizes is enough for the study of the effect of the spatial resolution.

*AR3.2:*

*We acknowledge that by only testing two grid sizes for the hydrological model we cannot claim to have done an exhausting analysis of the sensitivity of the model to the spatial resolution. As we have written to the first reviewer (RC1), we concede that the manuscript lacks a justification for the choice of discretization of both the geological voxel models and the hydrological models. We have added additional text to the method section to justify our approach and choices.*

*Furthermore, we have elaborated on the topic of choice of grid sizes for urban hydrogeological models in the discussion section during the revision of the manuscript.*

AC3.2:

In the material and methods section of the revised manuscript page 7, lines 169-179 the following text has been altered:

The effect of the geological configuration was analyzed by applying three geological models V0, V1, and V2 in otherwise identical  hydrological models.  The effect of spatial discretization was tested by using a coarse discretization relative to the urban subsurface infrastructure, with a horizontal grid discretization of 50 m, and a finer discretization close to the scale of the urban subsurface infrastructure, e.g., roads and trenches, with a horizontal grid discretization of 10 m. For both the geological and hydrological modeling tools a discretization in the order of 1-10 m becomes computationally challenging when the size of the model is large, resulting in e.g., millions of grid cells. For the hydrological modeling, multiple grid sizes were initially tested. The two grid sizes of 50 and 10 m were chosen based on a tradeoff between computation time and the number of grid cells for the model size and retaining geological detail. The effect of the geological configuration and spatial discretization  was analyzed based on the hydrological models' simulation of high-water levels, the 95$^{th}$ percentile, and particle tracking.

In the revised manuscript page 8, lines 222-209 the following text has been added:

anthropogenic structures in the uppermost subsurface and computational constraints. The choice of discretization for the voxel models with urban geology was guided by the experience from the study of Mielby & Henriksen (2020) and Mielby &

Sandersen (2017) and chosen to be 5x5x1 m to be able to represent the subsurface infrastructure trenches which are typically 1-3 m wide and 1-2 m in depth for this study area, while the Road trenches are around 10 m wide +/- and therefore 5 m is a good intermediate size. The horizontal discretization was thus larger than the dimensions of the trenches. A smaller discretization for a model at the city scale would have been computationally expensive in the hydrological model. Mielby & Sandersen (2017) argued that the discretization of the geological and hydrological model must meet the required detail, yet not exceed the computational capabilities. The two voxel models each had 22 million voxel grids.

In the revised manuscript page 27, line 566, the following sentences have been added to the beginning of the discussion section:

We tested the impact of urban geology for two discretizations; a coarse discretization and a finer discretization close to the scale of the urban subsurface infrastructure.

RC3.3:

The overall manuscript, including methods and results, is relevant and well-prepared and written. However, I have a few minor comments that could be into account in order to improve the quality of the manuscript and help the reader.

First of all, I noticed a lack of justification, especially in the Methods section. The authors did not always argue their assumptions :

- p5 l 118 : "… concrete pavement , which have an imperviousness of 75%" . How was this value estimated? Traditionally, this kind of surface is considered as totally impervious. But I acknowledge that it may be partially pervious. But that should be explained.

*AR3.3:*

*The quoted sentence says above 75% in the manuscript. The sentence is a description of the map in Figure 1b, which shows the imperviousness in percentage in 10 m grids. The data on imperviousness is from a raster map from the Danish Geodata Agency (2019). Buildings and pavements are as the reviewer points out normally considered 100% impervious. Yet, the map contains areas where the imperviousness is 75%. This can be places where a little area with vegetation is placed next to a building or a road. As suggested by the reviewer, we have specified this in the revised manuscript.*

AC3.3:

In the revised manuscript page 5, lines 126-128 the following text has been added:

asphalt, and concrete pavement, which have an imperviousness above 75 %. %. Buildings and pavements are normally considered 100% impervious. Yet, the map contains areas where the imperviousness is 75%. This can be areas where a little area with vegetation is placed next to a building or a road.

RC3.4:

-p7 l 183 : " … and additional data on soil material in the top 5 meters". As the modelling application is quite sensitive to the soil configuration, especially in the first meters, one can wonder where this "additional data" comes from! What kind of additional data? From drilling data? From infiltration tests?

*AR3.4:*

*As suggested by the reviewer, we have specified which data was used on soil material during the manuscript revision.*

AC3.4:

In the revised manuscript pages 8-9, lines 213-217, the following text has been added in the method section:

sources utilized for the geological models are presented in the supplementary material (Table S1). The additional data on soil material in V2 is a soil map (Jacobsen et al. 2022) and soil descriptions from shallow geotechnical boreholes (GEUS, 2019). The soil map by Jacobsen et al. (2022) is in 1:25000 resolution and is based on samples of soils every 200 m at 1 m depth. The soil descriptions from shallow geotechnical boreholes were derived by looking through non-digitalized documents in the Danish National well database.

RC3.5:

-P8 l 207-209 " the location of roads and pipes (…) were used as proxies for the presence of excavations and trenches" What is the relevance of this assumption? Did you assess this assumption? Did you compare this proxies methodology to real data? Is it valuable only in this study case or could it be transposed in any urban catchment?

*AR3.5:*

*We acknowledge that the quoted sentence is a vague formulation of the methodology of defining the extent of infrastructure in the geological voxel models. We have strived to make this clear in the revised manuscript.*

*The location of the roads and the pipes were retrieved from the road directory and the pipe owners, see table S1 for data sources. The extent of the excavations and trenches was based on national standards for profiles of road design and pipe trenches, see table S1 for sources. It was assumed that the design of the roads, railways, and trenches followed these standards.*

*To answer the reviewer's questions we find that this method of proxies for the presence of excavations and trenches is the best possible way unless the extent of the trenches is documented and stored in a central and digitalized archive. This is not the practice for this study area and we suspect it is rarely the case for other cities. Moreover, to answer the last question from the reviewer, the presented methodology can be transposed to other urban catchments.*

AC3.5:

In the revised manuscript page 10, lines 239-243, the following alterations have been made:

 The location of the roads and the pipes were retrieved from the national road directory and the pipe owners as GIS files, see table S1 for data sources. The extent of the excavations and trenches was based on national standards for profiles of road design and pipe trenches, see table S1 for sources. It was assumed that the design of the roads, railways, and trenches followed these standards.

RC3.6:

- P8 214-220 – Why the SHE model was chosen here? We can understand that it is the model used by the research team, but could the authors argue why this model is appropriate to do this study? Are there any equivalent modelling tools/methods that could have been considered for this type of modelling study? Is SHE model the only one that allows to achieve the objectives of this study?

*AR3.6:*

*We acknowledge that the manuscript lacks an argumentation for the choice of model code. In the revised version of the manuscript, we have elaborated on this at the beginning of section 3.2 Hydrological models. Generally, the section has been modified substantially after receiving the comments given to us by reviewer RC3. We have added an argumentation for the choice of model and a schematic figure of the hydrological model.*

AC3.6:

In the revised manuscript page 10-12, lines 250-288 the following text in section 3.2 Hydrological models have been altered a new figure have been added:

The study includes six hydrological models. The differences between the models were, as presented in Table 1, the application of two different horizontal discretizations and the three geological models used for the simulation of the subsurface processes. The hydrological models were based on the MIKE SHE code (Abbott et al., 1986 a,b). THE MIKE SHE code was chosen because of its ability to integrate the surface and subsurface processes dynamically, as well as its ability to include both overland and sewer drainage. Moreover, with this model code, the properties of the subsurface and the computational layers can be spatially distributed in both the horizontal and vertical planes. Other integrated hydrological models such as PARFLOW.CLM, MODFLOW 6, and HydroGeoSphere offer similar capabilities.

**Error! Reference source not found.** illustrates the common setup of the six hydrological models. The model components were overland flow, unsaturated zone flow, and saturated zone flow. The MIKE SHE models were coupled to a MIKE HYDRO model (DHI, 2017) for the simulation of groundwater seepage to the river bed and river discharge in the two creeks within the model domain.

[Figure]

**Figure 3. Illustration of the model setup for the hydrological models**

The overland flow was described by a finite difference approximation of the 2D Saint Venant equations for diffusive flow. The unsaturated zone flow was described by a simplified two-layer water balance approach assuming vertical flow and a conceptual formulation for actual evapotranspiration. This approach is primarily applicable to areas where the groundwater table is shallow and the actual evaporation rate is close to the potential rate (Butts and Graham, 2005), which is the case for the study site. The saturated zone flow was described by the governing equation for 3D saturated flow based on Darcy's law. Subsurface drainage was included as a sink term and depended on the groundwater level, depth of the drains, and a time constant. Detailed descriptions of the components can be found in DHI (2017, 2020).

to the 2D Saint Venant equations ~~for diffusive flow. Overland flow is generated when precipitation is higher than the infiltration capacity due to either high groundwater levels or ponding at the surface. Reduced infiltration capacity was specified in paved areas, and drainage of ponded water was specified by the imperviousness of the land cover. (2) Unsaturated flow, described by a simplified two-layer water balance approach assuming vertical flow and a conceptual formulation for actual evapotranspiration. This approach is primarily applicable to areas where the groundwater table is shallow and the actual evaporation rate is close to the potential rate (Butts and Graham, 2005), which is the case for the study site. (3) Saturated flow, described by the governing equation for 3D saturated flow based on Darcy's law. Subsurface drainage is included as a sink term in the equation and depends on the groundwater level, depth of the drain, and a time constant. Detailed descriptions of the components can be found in DHI (2020).~~

The computational  steps were automatically controlled to secure accurate water balances. The maximum time steps  were: 0.5 hours for overland flow, 6 hours for unsaturated flow, and 12 hours for groundwater flow.  The MIKE HYDRO models used  the kinematic wave approximation  with a fixed time step of 10 minutes.

RC3.7:
- p9 l 245. What is this surface-subsurface leakage coefficient? A parameter of the SHE model? Does it take into account the leakage in pipes, or only the leakage from surface-subsurface? How could it be estimated?

*AR3.7:*
*The surface-subsurface leakage coefficient is a model parameter in MIKE SHE. It reduces the infiltration from the surface to the subsurface at paved surfaces as well as the seepage from the subsurface to the surface. In the model, it is applied to the areas where the imperviousness is above 50%.*
*Surface-subsurface leakage coefficient does not account for leakage in pipes. Leakage in the pipes was modeled separately and it was assumed that leakage only occurs in the sewer pipes. The leakage was modeled by representing the sewer network as subsurface drainpipes and assuming that the parameter was spatially uniform across the pipe network.*
*We have strived to make this clear in the revised manuscript.*

AC3.7:
In the revised manuscript page 13, lines 305-310 the text has been changed to:
detention storage. Moreover, it was used as a linear scaling fraction for the surface-subsurface leakage coefficient. The surface-subsurface leakage coefficient reduces the infiltration from the surface to the subsurface at paved surfaces as well as the seepage from the subsurface to the surface. In the model, it is applied to the areas where the paved area fraction is above 0.5 and it was given the value 6x10-7 s-1 in all cells and then scaled by the paved area fraction. The scaling of the surface-subsurface leakage coefficient by the paved area fraction is referred to as the effective leakage coefficient (DHI, 2020).

RC3.8:

Then, the methods section could have been improved with a graphical scheme helping the reader to understand the chosen parametrizations. This is especially needed in the 3.2.2 paragraph, because the list of the presentation of the parametrization and boundary conditions is quite long, and a scheme would be more efficient and more easy for the reader.

*AR3.8:*

*The parameterization is indeed complicated. Some parameters are defined from data, some were specified from past model experience and some were subject to calibration as described in section 3.3. To enhance the readability of the parameterization we have added a graphical scheme of the hydrological model set up for the six hydrological models.*

AC3.8:

Please see the changes present under AC3.6.

RC3.9:

Finally, I have a short comment about one element of discussion : l 535-543. The sewers renovation could be a way to reduce the soil-sewer interactions and the infiltration of groundwater in sewers. As discussed by the authors, the preferential flow paths would still be present in the pipe trenches. However, I wonder if having a full renovated sewer system is not an utopy… To my opinion, there will still be some defects in the sewer system and then, as the preferential flow in the trenches remains present, the water will always find a way to penetrate in the sewers. I have the impression that this type of sewer renovation (or "non leaking pipes assumption") is only a "modelling dream"; I am not sure it would be feasible in reality.. (especially in a economical point of view). I would appreciate that the authors re-consider this paragraph.

*AR3.9:*

*We agree that it is probably not realistic to install a leakage-free pipe network. As the reviewer correctly states, the preferential flow in the trenches remains present, and the water will always find a way to the leaking sewers. In the discussion of the revised manuscript, we speculate on the possible impact of the renovation of sewers and we will expand on this topic in the revised manuscript. We have rephrased this section of the discussion in the revised manuscript.*

AC3.9: In the revised manuscript pages 28-29, lines 616-625 the text has been changed:

~~It was assumed that the entire sewer system was leaky and thus acted as a drainage system as well. In consequence, drains in the saturated zone were by far the most dominant sink to the shallow groundwater, Figure 12 and Figure 13. Although this assumption may be on the extreme side, groundwater seeping into the sewers is a common problem and leads to excessive water treatment in areas with shallow groundwater. On the other hand, in cases where such sewers are renovated, the water table may raise and trigger water seeping into basements or a periodical groundwater table above the terrain. The six model simulations would most likely be different if the sewer system was not leaking. The preferential flow paths would still be present, but the water in those locations would not be drained.~~

It was assumed that the entire sewer system was leaky and thus acted as a drainage system as well. A consequence of this was that the drains in the saturated zone were by far is the most dominant sink to the shallow groundwater, see Figure 12 and Figure 13. Although this assumption about leaky sewers may not be correct in all areas of the model, groundwater seeping into the sewers is a common problem and leads to treatment of excessive quantities of water in areas with shallow groundwater (Bhaskar et al., 2015; Rasmussen et al., 2022). If the pipe network in the model area is partly renovated, the water table may rise and reach unrenovated household drains that feed into the central storm or sewer network.

Table 3